# Patient-specific models link neurotransmitter receptor mechanisms with motor and visuospatial axes of Parkinson's disease

Ahmed Faraz Khan[1,2,3], Quadri Adewale[1,2,3], Sue-Jin Lin [1,2,3], Tobias R. Baumeister[1,2,3], Yashar Zeighami[1,4], Felix Carbonell[5], Nicola Palomero-Gallagher [6,7,8] & Yasser Iturria-Medina [1,2,3] ✉

Parkinson's disease involves multiple neurotransmitter systems beyond the classical dopaminergic circuit, but their influence on structural and functional alterations is not well understood. Here, we use patient-specific causal brain modeling to identify latent neurotransmitter receptor-mediated mechanisms contributing to Parkinson's disease progression. Combining the spatial distribution of 15 receptors from post-mortem autoradiography with 6 neuroimaging-derived pathological factors, we detect a diverse set of receptors influencing gray matter atrophy, functional activity dysregulation, microstructural degeneration, and dendrite and dopaminergic transporter loss. Inter-individual variability in receptor mechanisms correlates with symptom severity along two distinct axes, representing motor and psycho-motor symptoms with large GABAergic and glutamatergic contributions, and cholinergically-dominant visuospatial, psychiatric and memory dysfunction. Our work demonstrates that receptor architecture helps explain multi-factorial brain re-organization, and suggests that distinct, co-existing receptor-mediated processes underlie Parkinson's disease.

Parkinson's disease (PD) is primarily associated with a nigrostriatal dopamine deficit resulting in the characteristic motor symptoms of tremor, rigidity, and bradykinesia. However, the involvement of other brain circuits is now widely recognized[1], and the majority of patients also present numerous non-motor symptoms such as dementia, depression, sleep disorders, or apathy[2]. For this multi-system disease with significant inter-patient heterogeneity in pathology, symptoms and treatment response[3–5] consistent links between genetic, neuropathological and clinical subtypes remain elusive[6]. With no cure[7], symptomatic pharmacological treatment (e.g., levodopa) is at best partially effective[8] and may result in undesired side effects with chronic administration[9]. Given that diagnostic accuracy in untreated or medication non-responder PD patients is as low as 26%[10], an improved understanding of biological mechanisms and potential therapeutic targets underlying pathological and symptomatic heterogeneity is imperative to bridging the treatment gap in PD[11–13].

[1]Department of Neurology and Neurosurgery, Montreal Neurological Institute, McGill University, Montreal, QC, Canada. [2]McConnell Brain Imaging Center, Montreal Neurological Institute, Montreal, QC, Canada. [3]Ludmer Centre for Neuroinformatics & Mental Health, Montreal, QC, Canada. [4]Douglas Research Centre, Department of Psychiatry, McGill University, Montreal, QC, Canada. [5]Biospective Inc, Montreal, QC, Canada. [6]Institute of Neuroscience and Medicine (INM-1), Research Centre Jülich, Jülich, Germany. [7]Cécile and Oskar Vogt Institute of Brain Research, Medical Faculty, Heinrich-Heine University, Düsseldorf, Germany. [8]Department of Psychiatry, Psychotherapy, and Psychosomatics, Medical Faculty, RWTH Aachen, and JARA - Translational Brain Medicine, Aachen, Germany. ✉e-mail: yasser.iturriamedina@mcgill.ca

Neurotransmission underlies many disease-related mechanisms as well as pharmacological response[8,14]. Regional variability in neurotransmitter receptor gene expression correlates with altered macroscopic interactions such as neurovascular[15] and structural-functional decoupling[16]. Multiple non-dopaminergic nuclei are affected in PD[17,18], with specific neurotransmitter systems linked to symptoms such as cholinergic freezing of gait and dementia[19], serotonergic depression and tremor[20], and adrenergic postural symptoms[21]. The dual-syndrome hypothesis of PD[18] proposes a dichotomy between dopamine-mediated fronto-striatal executive impairment and a cholinergically-mediated prodromal visuospatial dementia. To better characterize the role of neurotransmission in mediating neurodegenerative brain reorganization, an integrative model linking multiple receptor systems, macroscopic brain reorganization and clinical symptoms would be essential. However, we are limited by the absence of whole-brain spatial distribution maps of neurotransmitter receptors in PD patients[8].

On the other hand, neuroimaging supports the multi-factorial and heterogeneous view of PD[22]. Various modalities are routinely used to support differential diagnosis[11,23,24] and evaluate treatment effects[25]. Multi-modal modeling of neuroimaging alterations can elucidate the temporal ordering, disease trajectories, and interactions of various pathologies in neurodegeneration[26,27], and link these macroscopic observations with underlying genetic and transcriptomic determinants[28]. Multifactorial causal modeling (MCM) is a mechanistic modeling approach that is able to identify contributions of interacting factors to longitudinal changes[29], which can be used in a personalized medicine context to design optimal therapeutic interventions[30]. Combining multi-modal neuroimaging with spatial distribution templates of 15 neurotransmitter receptors from post-mortem autoradiography[31] in an MCM-based approach significantly improved the explanation of degenerative changes in individual patients' neuroimaging data, and linked specific receptor-pathology interactions to clinical symptoms in Alzheimer's disease (AD)[32]. Furthermore, this approach was able to estimate individualized receptor alterations based on inter-subject differences in receptor-neuroimaging interactions.

Here, we extend previous molecular-phenotypic PD characterizations in four fundamental ways: (i) by combining spatial distribution maps of 15 key neurotransmitter receptors derived from post-mortem autoradiography[31] with longitudinal neuroimaging data in a personalized modeling framework to infer the individualized importance of various receptor-mediated interactions ($N = 71$, PPMI data), (ii) by demonstrating the improved ability of receptor-enriched multifactorial causal modeling (re-MCM) to explain imaging-measured neurodegeneration and identify consistent mechanistic changes across patients, (iii) by characterizing inter-patient heterogeneity, specifically linking receptor-based mechanistic alterations to two main axes of motor, cognitive and psychiatric symptoms, (iv) quantitatively mapping brain regions with high receptor influence on PD neurodegeneration.

## Results

### Model-based approach to inferring personalized neurotransmitter receptor alterations

To characterize neurotransmitter receptor contributions to the multifaceted neurodegenerative processes of PD, we fit receptor-informed individualized generative computational models to the longitudinal alterations of 6 biological factors. Each biological factor is associated with neurodegeneration in PD, namely atrophy, dysregulated functional activity, dopaminergic deficiency, directed and microstructural damage, and dendrite loss, represented by the neuroimaging-derived measures of gray matter density (GM), fractional amplitude of low frequency fluctuations (fALFF), dopamine transporter SPECT (DAT-SPECT), fractional anisotropy (FA), mean diffusivity (MD), and t1/t2

ratio[33,34]. Neuroimaging data was acquired over multiple imaging scans for $N = 71$ PD patients (PPMI data, Methods: Data description and processing). In addition, regional densities for 15 neurotransmitter receptors (from glutamatergic, GABAergic, cholinergic, adrenergic, serotonergic, and dopaminergic families) were derived from averaged templates (Methods: Data description and processing: Receptor densities and brain parcellation), and anatomical connectivity was estimated from the high-resolution Human Connectome Project template (HCP-1065; Methods: Anatomical connectivity estimation).

The neurotransmitter receptor-enriched multifactorial causal model (re-MCM; Fig. 1) decomposes the spatiotemporal evolution of pathology of multiple biological factors into localized receptor- and network-mediated effects (Fig. 1a). Model parameters explicitly represent distinct biological mechanisms, namely (i) direct and (ii) receptor-mediated pairwise interactions between imaging-derived biological factors (dopaminergic deficiency, functional activity, microstructural damage, dendrite density, and atrophy), iii) effects of local neurotransmitter receptor densities on factor-specific longitudinal deterioration, and (iv) spreading of pathology to and from anatomically connected regions. Notice that, in the absence of true personalized longitudinal receptor imaging, model weights of specific receptor-mediated biological mechanisms compensate to fit individualized trajectories of neurodegeneration. Thus, inter-subject variability in model weights serves as a proxy for the corresponding receptor densities or receptor-pathology interactions. Specifically, (i) the improvement of model fit by the inclusion of healthy aged receptor templates validates their application to this clinical population, (ii) biological mechanisms that are statistically stable across subjects represent mechanistic pathways shared by all PD patients in our cohort, (iii) inter-patient co-variability between biological mechanisms and clinical symptoms represents overlapping disease processes (Fig. 1b), and (iv) inter-region variability in the model fit improvement due to receptor templates can identify regions differentially affected by neurotransmitter receptor alterations in PD (Fig. 1c).

### Neurotransmitter receptor maps significantly improve the explanation of multi-factorial brain reorganization in PD

Before proceeding to identify relevant model-derived biological mechanisms in PD, we first aimed to validate that re-MCM robustly fits patient-specific neuroimaging data. For each of the 6 biological factors and all subjects ($N = 71$), we calculated the coefficient of determination ($R^2$) as a measure of the data variance explained. On average, re-MCM explained 74% ± 18% of the variance in rate of pathology accumulation (Fig. 2a), although model fit varied by biological factor, with neural activity dysfunction (fALFF; 81% ± 11%), dopaminergic degeneration (DAT-SPECT; 80% ± 13%) and dendrite loss (t1/t2 ratio; 80% ± 12%) being explained better than gray matter atrophy (GM; 58% ± 14%), or microstructural damage (MD; 70% ± 14%, and FA; 0.74 ± 0.13). For validation, we repeated the model-fitting without receptor-pathology interactions or direct local receptor density effects. On average, neuroimaging-only models without receptor data explained 52% ± 20% of the variance in neuroimaging rate of change (Fig. 2b), and the inclusion of receptor templates improves the data variance explained by 42.3%. Dopaminergic loss (DAT-SPECT) was the least improved by the addition of receptor maps, with imaging-only models explaining 60% ± 17%, a drop of 20% of variance on average compared to the full re-MCM. On the other hand, gray matter atrophy (GM: 22% ± 17% variance explained without receptor maps) was the most reliant on receptor data. While DAT-SPECT scans themselves already image the density of presynaptic dopaminergic transporters, gray matter atrophy models benefit more from regional differentiation based on receptor expression.

Figure 2c presents the improvement in each participant's model fit due to receptor mechanisms, compared to the restricted, neuroimaging-only models. Accounting for the increased model size

**a  Individualized causal modeling of interacting pathological factors**

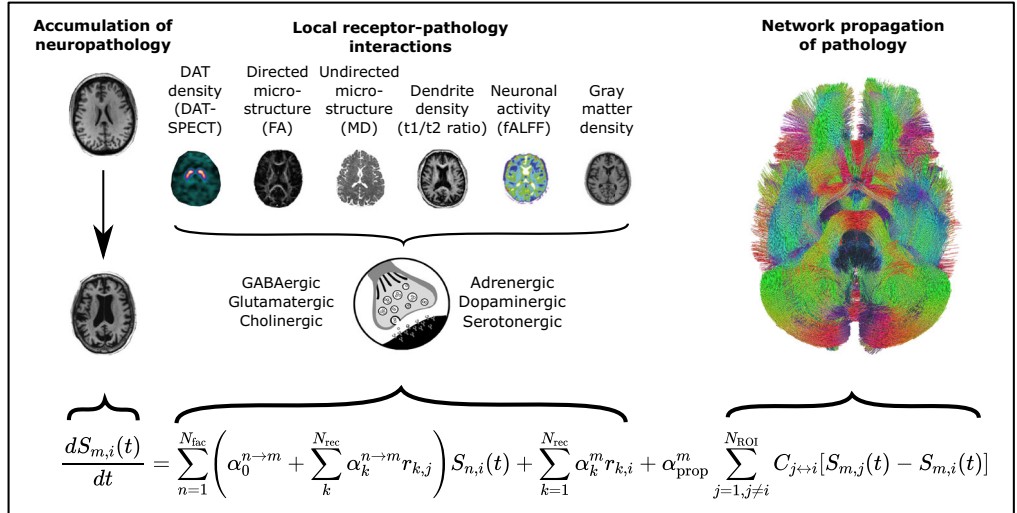

$$\frac{dS_{m,i}(t)}{dt} = \sum_{n=1}^{N_{fac}} \left( \alpha_0^{n \to m} + \sum_{k}^{N_{rec}} \alpha_k^{n \to m} r_{k,j} \right) S_{n,i}(t) + \sum_{k=1}^{N_{rec}} \alpha_k^m r_{k,i} + \alpha_{prop}^m \sum_{j=1, j \neq i}^{N_{ROI}} C_{j \leftrightarrow i} [S_{m,j}(t) - S_{m,i}(t)]$$

**b  Population-level association of model-derived mechanisms and clinical symptoms**

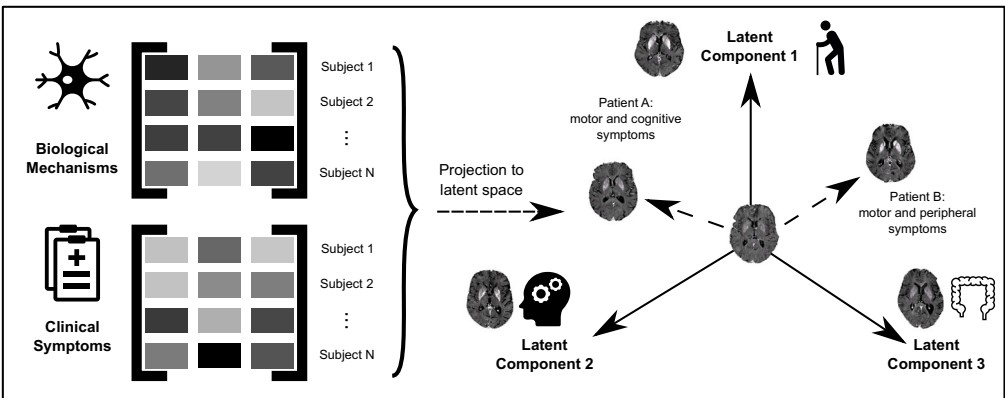

**c  Population-level estimation of regional receptor influence**

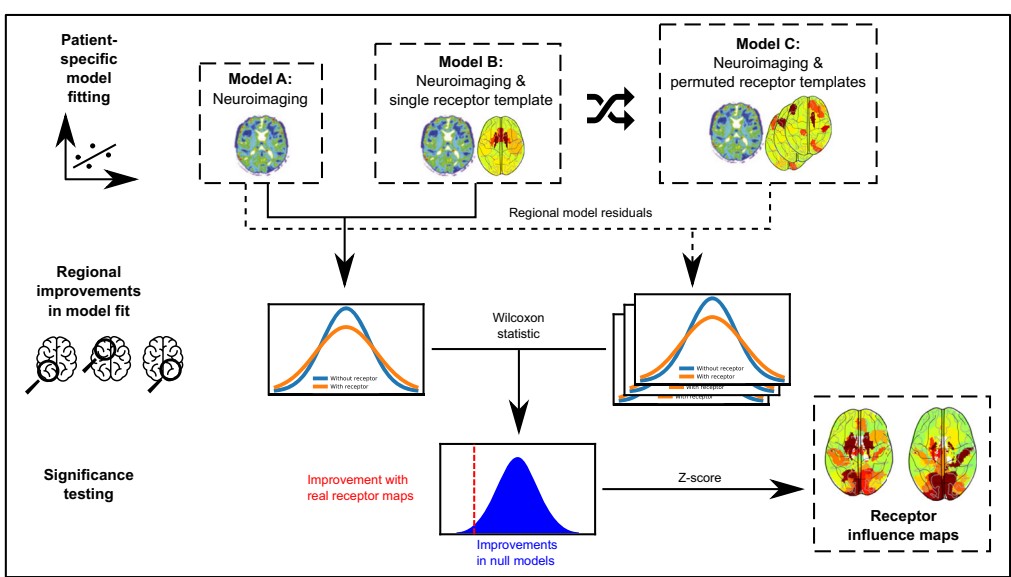

from 8 to 113 parameters, the F-statistics of 80.3% (MD) to 100% (DAT-SPECT) of patients is significant ($p < 0.05$ red dotted line in Fig. 2c). We then performed a permutation test for the significance of the informativeness of receptor maps, by randomly shuffling each receptor map across brain regions 1000 times and fitting the re-MCM with each set of permuted maps.

The resulting distribution of model fit ($R^2$) was used to calculate significance levels for re-MCM with true receptor data from Fig. 2a. For each biological factor, we plotted the number of subjects with significantly better model fit ($p < 0.05$) compared to the null distribution in Fig. 2d. Notably, nearly all patients' biological factor models are significantly improved by the inclusion of receptor maps, except for

**Fig. 1 | Neurotransmitter receptor-enriched multifactorial causal modeling. a** Each patient's longitudinal pathological progression is decomposed into local effects due to: (i) direct influence of every imaging-derived biological factor (e.g., atrophy on resting state functional activity), (ii) receptor density distribution (e.g., $D_1$ receptor density on DAT loss), and (iii) receptor-pathology interactions (e.g., $D_1$ receptors × DAT interactions on functional activity), in addition to (iv) network-mediated inter-region propagation. Combining this data across ($N_{ROI} = 95$) brain regions and multiple visits results in a multivariate regression problem to identify the patient-specific parameters {α}. **b** Decomposing the covariance matrix of patients' model-derived biological mechanism weights and clinical scores

(specifically, the rates of decline of composite clinical scores; Methods: Clinical scores) identifies multivariate axes of receptor-factor interactions that are robustly correlated with the severity of combinations of clinical symptoms in PD (Methods: Biological parameters and relationship with cognition). **c** The regional contributions of receptor interactions to neurobiological changes are estimated by a feature importance analysis. We fit individualized models for every biological factor with and without each receptor map and performed permutation tests on the improvement in regional model residuals due to the inclusion of receptor maps. The resulting improvements are the significant regional influence of receptors on each target biological factor model.

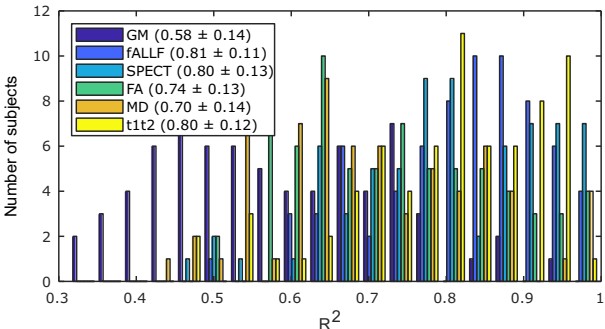

**a Data variance explained by receptor-imaging interaction model**

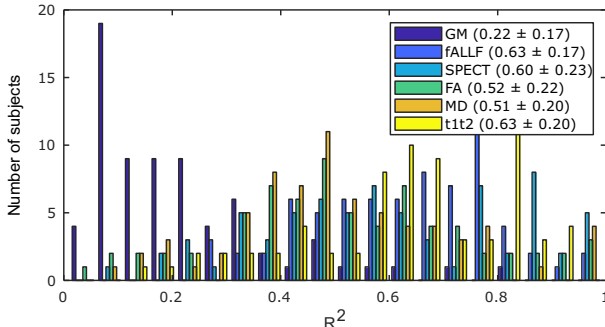

**b Data variance explained by neuroimaging-only model**

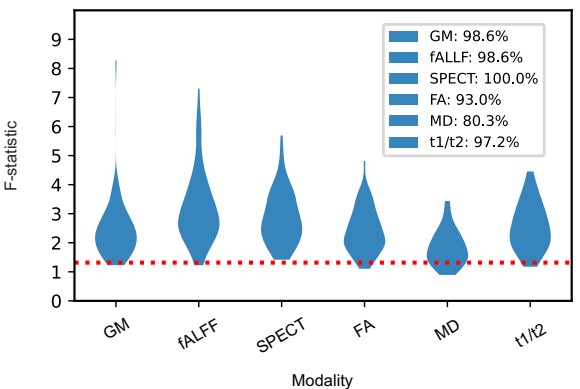

**c Receptor templates improve most subjects' neuroimaging models**

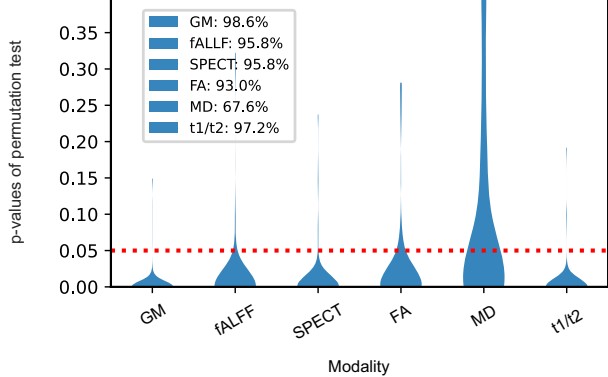

**d Receptor templates outperform null maps in most subjects**

**Fig. 2 | Contribution of receptor distributions to explaining multimodal brain reorganization in PD.** Pathological factors are quantified by 6 neuroimaging-derived metrics: gray matter density (GM), neuronal activity (fractional amplitude of low frequency fluctuations; fALFF), dopamine transporter density (DAT) from SPECT, directed microstructure (fractional anisotropy; FA), undirected microstructure (mean diffusivity; MD), and dendrite density (t1/t2 ratio). The improvement in modeling the accumulation of pathology was evaluated in terms of (i) the additional explanatory power due to receptor information, and (ii) the significance of true receptor maps compared to null distributions. The histograms show the distribution of the coefficient of determination ($R^2$) of $N = 71$ individual models of longitudinal neuroimaging changes including (**a**) and excluding (**b**) receptor predictors. Notably, including receptor terms improves model fit for all biological

factors, although to varying extents. **c** Subject-wise *F*-tests between models with and without receptor maps (113 and 8 parameters, respectively) show proportions of subjects for whom the F-statistic is above the critical threshold (red dotted line). This critical threshold corresponds to a statistically significant ($P < 0.05$) improvement due to the receptor terms in the re-MCM model, accounting for the increase in adjustable model parameters. Furthermore, to validate the benefit of the receptor templates over randomized null maps, re-MCM models were fit with 1000 spatially permuted receptor maps for each subject. The *p* value of the model fit ($R^2$) using true receptor templates compared to the distribution of $R^2$ of models using randomized templates was calculated for each subject. **d** Proportion of subjects for whom the true receptor maps resulted in a statistically significant improvement in model fit ($P < 0.05$; red dotted line).

undirected microstructural damage (MD; 67.6% or 48 subjects). Across all participants, Fisher's method gives $\chi^2$ statistics in the range of $800 < \chi^2 < 2300$ (depending on the biological factor), corresponding to a near-zero combined *P* value. These analyses validate the use of averaged receptor templates in patient-specific PD models.

## Identifying stable neurobiological mechanisms and receptor-pathology interactions in PD

We proceeded to identify biological mechanisms consistently involved in structural, functional, and dopaminergic brain alterations in PD. For

this, 99% confidence intervals for each re-MCM parameter across all patients were calculated and used to identify stable predictors. Since all predictors were standardized before data fitting, model weights are the relative effect sizes of different biological mechanisms on the rate of change of their target biological factor over the course of PD progression. Specifically, these neurobiological mechanisms are (i) direct effects of local pathology, (ii) direct effects of local receptor densities, (iii) local receptor-pathology interactions, and (iv) network propagation of pathology (Methods: Receptor-Enriched Multifactorial Causal Model).

**a   Multiple receptor mechanisms are important predictors of pathological accumulation**

**b   Pathological factors interact in complex pathways**

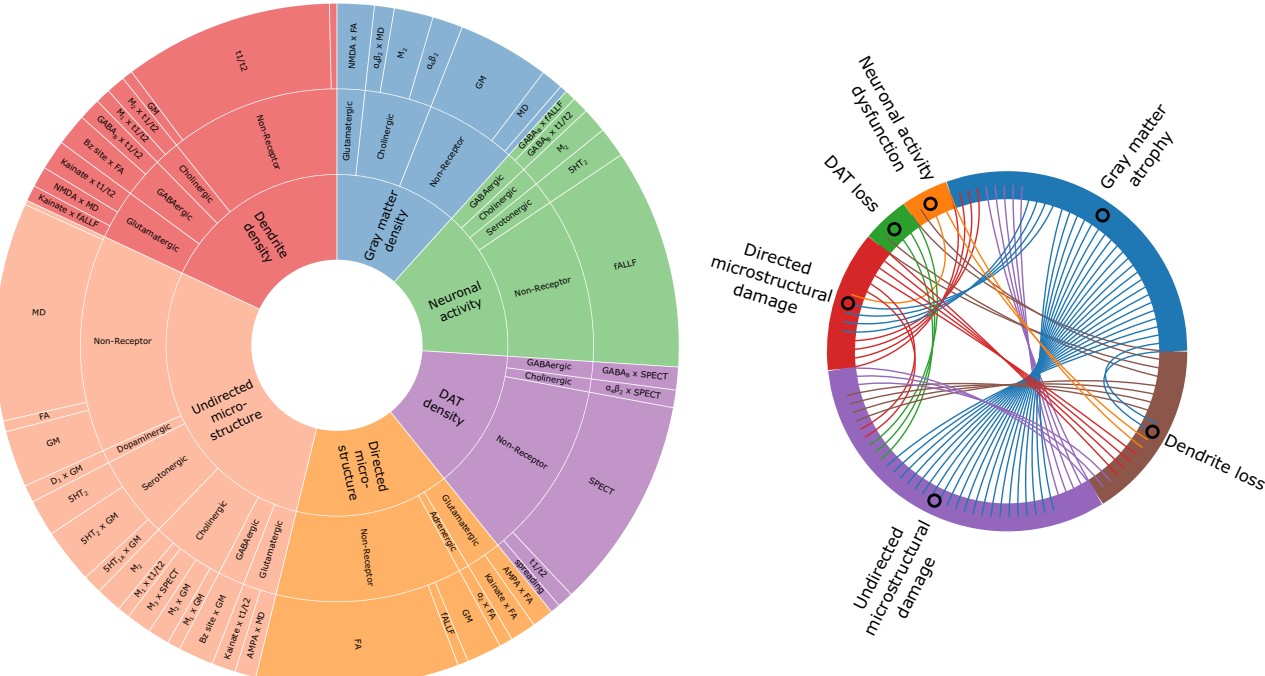

**Fig. 3 | Receptor-mediated interactions explaining longitudinal neurodegeneration in PD. a** Statistically stable biological mechanisms in PD show significant receptor-mediated contributions. The angle of each outer sector is proportional to the mean weight of each stable (99% confidence interval) re-MCM model weight across the PD patients. The inner sectors represent the 6 modeled biological factors. Within each factor, the intermediate sectors represent the neurotransmitter

system involved, while the outer sector consists of the specific two-way receptor-pathology interactions or direct predictor terms in the model. Notably, biological factors may appear as both model predictors (outer sector) and targets (inner sector). **b** Effect size (number of chords) of statistically stable interactions between any pair of biological factors modeled in PD.

Figure 3a shows the relative effective sizes of stable biological mechanisms. The most influential stable predictors of each biological factor's rate of change are the direct effects of local alterations to the same modality. Propagation of pathology along the structural connectome is also a minor yet stable predictor for all data modalities except functional activity (fALFF) and directed microstructural damage (FA), with a much lower effect than the local evolution of neurodegeneration. Notably, from Fig. 3b, functional brain alterations (fALFF) do not appear to drive structural alterations (GM and MD), instead interacting bidirectionally with dendritic density (t1/t2).

Nevertheless, local interactions between imaging-based biological factors, whether direct or receptor-mediated, constitute a significant driver of PD neurodegeneration in all cases, and form a complex network with potentially bidirectional influences (Fig. 3b). While comparatively smaller for functional activity, dopaminergic transporter density and directed microstructural integrity (FA), receptor-mediated interactions constitute approximately half the model effects for gray matter atrophy (GM), overall microstructural integrity (MD) and dendrite density (t1/t2).

We observed that a relatively sparse set of receptors is involved in stable interactions for each biological factor (Fig. 4). The muscarinic $M_2$ and nicotinic $\alpha_4\beta_2$ receptors contribute significantly to gray matter atrophy, neuronal activity dysfunction, and dopaminergic loss. The Bz site is also prominently associated with neuronal activity dysfunction and dopaminergic loss. The serotonergic $5HT_2$ receptor is involved in functional and undirected microstructural alterations, while glutamatergic effects are marked by NMDA affecting gray matter atrophy, AMPA and kainate affecting directed microstructure and kainate affecting dendrite density, respectively.

Generally, the dopaminergic, cholinergic, serotonergic, glutamatergic and GABAergic systems broadly affect (micro-)structural

alterations (GM, MD and t1/t2). Serotonergic mechanisms are most associated with undirected microstructural alterations (MD), and secondarily dysfunctional neural activity (fALFF). Cholinergic receptors are prominent predictors of atrophy, microstructural damage and loss of dendrites (GM, MD and t1/t2), with minor influence on functional activity and dopaminergic transporter density. Glutamatergic receptors have a moderate influence across structural modalities (GM, MD, FA and t1/t2). GABAergic influence is minor yet stable across functional (fALFF and SPECT) and (micro-)structural (MD and t1/t2) modalities. Adrenergic and dopaminergic receptors are the least involved in stable neurobiological mechanisms, with $\alpha_2$ adrenergic receptor modulating directed microstructural damage (FA), and the $D_1$ dopaminergic receptors mediating the effect of atrophy on microstructure (MD).

For atrophy (GM), functional activity (fALFF) and microstructure (MD) models, the direct effects of specific receptor density maps reflect local susceptibility to neurodegeneration. The densities of the muscarinic $M_2$ and nicotinic $\alpha_4\beta_2$ cholinergic receptors help explain inter-region variability in the rate of gray matter atrophy, while $M_2$ and the serotonergic $5HT_2$ receptor densities are stable predictors of both altered activity (fALFF) and microstructural damage (MD).

### Two axes of receptor-pathology alterations underlie clinical symptoms in PD

To link model-derived receptor-mediated neurobiological mechanisms with clinical presentation in PD, we identified shared axes of covariance between re-MCM-derived biological mechanisms and motor, non-motor, cognitive and psychiatric symptoms (Methods: Clinical scores). Partial least squares (PLS) regression using singular value decomposition (SVD) across all patients ($N = 71$) was used to identify multivariate and overlapping relationships between

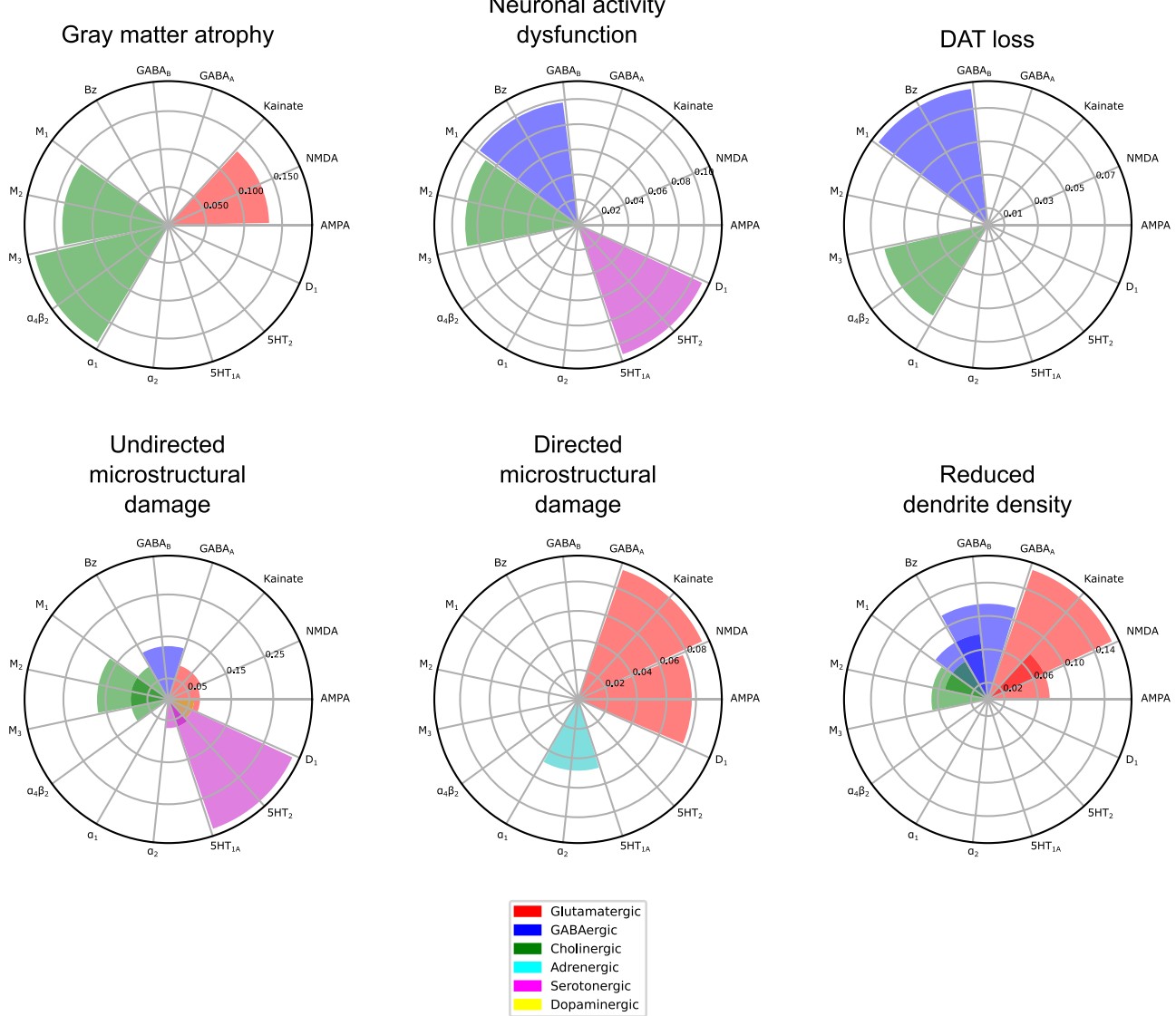

**Fig. 4 | Receptors mediating degenerative alterations to different macroscopic biological factors in PD.** The combined statistically stable model effects of each receptor type on each biological factor are shown. The muscarinic $M_2$ and nicotinic $\alpha_4\beta_2$ receptors contribute significantly to gray matter density, neuronal activity and dopamine transporter alterations. The Bz site is prominently associated with activity and dopamine transporter alterations. The serotonergic $5HT_2$ receptor is involved in functional and microstructural (MD) alterations, while glutamatergic effects are marked by NMDA affecting gray matter atrophy, AMPA and kainate affecting directed microstructural damage (FA) and kainate affecting dendrite density (t1/t2), respectively. Notably, the $D_1$ receptor distribution is relatively homogeneous and not marginally informative in the presence of DAT imaging.

identified biological parameters and clinical symptoms (Methods: Covariance of biological mechanisms with clinical symptoms) via projections to a latent space. Two latent components were relevant based on permutation tests, explaining 48.4% ($P = 0.001$, FWE-corrected) and 13.2% ($P = 0.069$, FWE-corrected) of the population co-variance, respectively. Projections of biological mechanisms and clinical scores to these components show moderate to high correlations of $r = 0.70$ ($P = 3.11 \times 10^{-11}$; Fig. 5a) and 0.86 ($P = 3.75 \times 10^{-21}$; Fig. 5b).

Interestingly, the first component (primary axis; Fig. 5c) largely corresponds to variance of the MDS-UPDRS Parts 1–3 scores (composed of cognitive, psychiatric and motor aspects of daily living, as well as a motor exam), and SDM (assessing attention, perceptual speed, motor speed, and visual scanning[35]). On the other hand, the second component (secondary axis; Fig. 5d) is associated with the BJLOT (visuospatial judgment), LNS (working memory), STAIAD

(anxiety) and the GDS (depression in older adults). The statistically stable biological mechanisms contributing to each axis are summarized in Fig. 6. Both components show that inter-subject symptom variability is associated with multiple receptor-mediated biological mechanisms and neuropathological changes. The primary axis is largely driven by GABAergic alterations (explaining 5.97% of the total covariance via this component), although glutamatergic (4.85%), cholinergic (4.77%), and serotonergic (3.77%) alterations are also prominent. The secondary axis is instead associated primarily with cholinergic alterations (1.74%), although GABAergic (1.24%) and glutamatergic (1.19%) alterations also play a role.

While the local (regional) evolution of pathology in each considered biological factor and its network propagation are prominent stable predictors of PD neurodegeneration (Fig. 3), the influence of these mechanisms does not co-vary prominently with symptom severity. Instead, we find a broad array of receptors with clinical effects

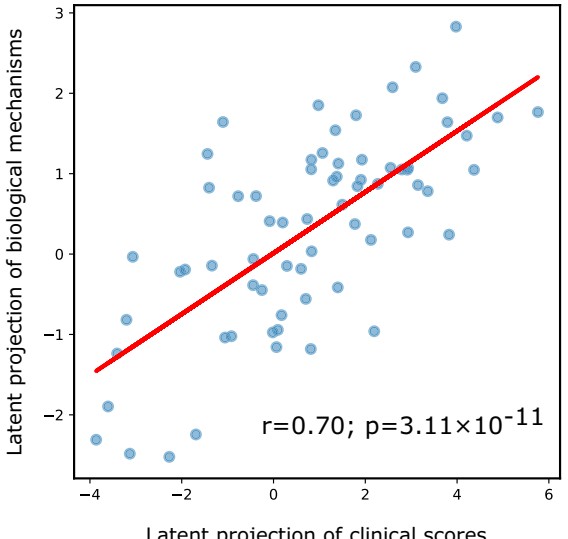

**a** Correlation between biological mechanisms and symptom severity along the primary axis

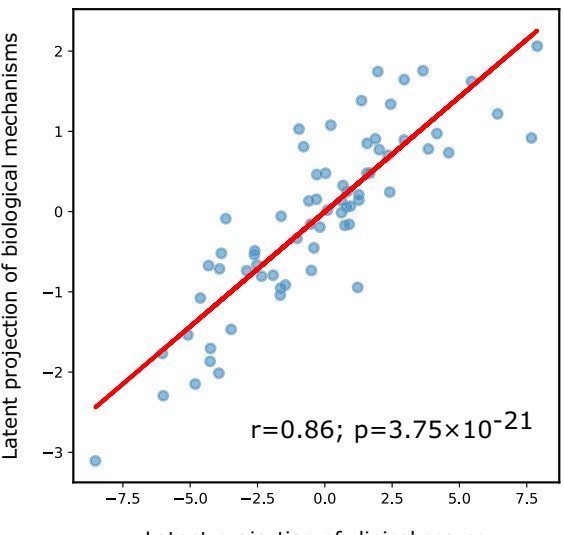

**b** Correlation between biological mechanisms and symptom severity along the secondary axis

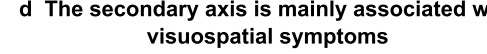

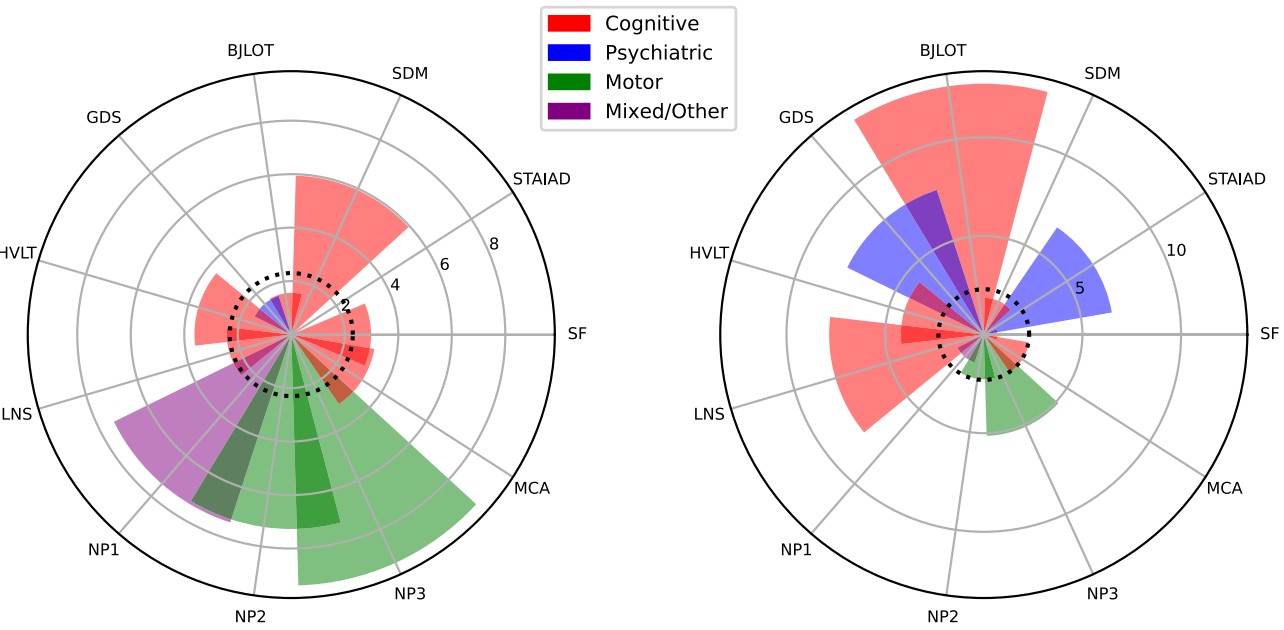

**c** The primary axis is mainly associated with motor symptoms

**d** The secondary axis is mainly associated with visuospatial symptoms

**Fig. 5 | Two axes of covariance between biological mechanisms and symptom severity in PD. a** Based on a permutation analysis, two latent SVD components were significant or near-significant, explaining 48.4% ($P = 0.001$, FWE-corrected) and 13.2% ($P = 0.069$, FWE-corrected) of the covariance respectively. **a**, **b** High correlations of $r = 0.70$ ($P = 3.11 \times 10^{-11}$) and 0.86 ($P = 3.75 \times 10^{-21}$), between the projections of statistically stable (based on 95% confidence intervals from bootstrapping) biological mechanisms and rates of clinical decline onto the latent components were observed. **c**, **d** Bootstrap ratios of each clinical assessment to the two latent components, providing a relative ranking of motor, nom-motor, psychiatric and cognitive domains. These saliences are proportional to the contribution of each term relative to every other term, for example showing that MDS-UPDRS scores, SDM and HVLT scores are the top contributors to the primary axis. Details about specific scores can be found in Methods: Clinical scores.

along both latent axes, as shown in Fig. 5. For example, the mainly motor symptoms of the primary axis are associated with inter-subject variability in glutamatergic and GABAergic interactions affecting microstructural integrity (MD and FA) and dendrite density (t1/t2). In contrast, the visuospatial, psychiatric and memory dysfunction of the secondary axis is associated more with inter-subject variability in cholinergic interactions affecting microstructure (MD) and dendritic density (t1/2), as well as changes to GM density.

## Mapping receptor influence in PD

Finally, we inferred the degree of receptor influence on multi-modal PD neurodegeneration at different brain regions, by identifying brain regions where the inclusion of a specific receptor predictor consistently improves the explanation of a particular type of neuropathology across all subjects. For each receptor, we fit individualized, single receptor-enriched models, and compared their ability to explain the accumulation of pathology at each brain region with restricted,

**a  Biological mechanisms contributing to the primary axis**

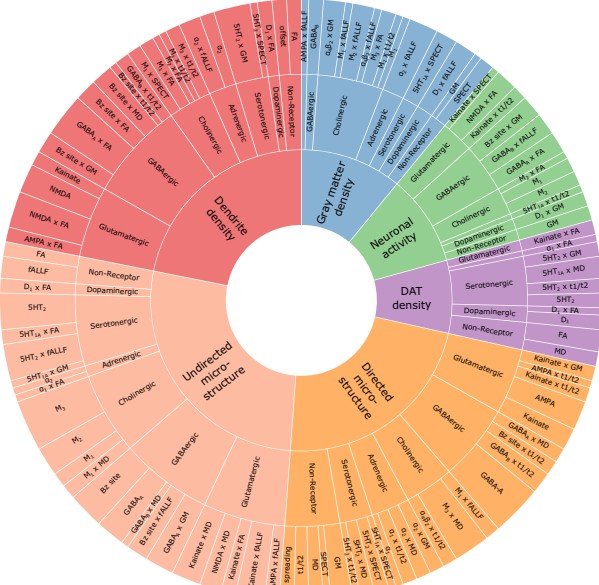

**b  Biological mechanisms contributing to the secondary axis**

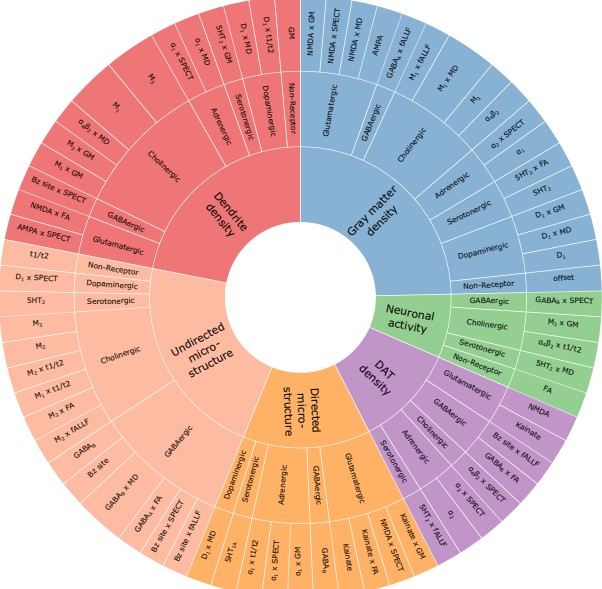

**Fig. 6 | Distinct combinations of receptor-mediated interactions are associated with the two axes of clinical symptoms.** Biological mechanisms correlated with clinical severity in PD via the (**a**) motor/psychomotor and (**b**) visuospatial/memory/psychiatric axes are plotted. Representing the effects of receptor densities, local pathology, receptor-pathology interactions, and network propagation of pathological factors, combinations of patient-specific mechanisms co-vary with specific clinical symptoms. Sector colors represent the output pathological factor of each model, named in the inner (central) sectors. For each mechanism, the angle is proportional to the percentage of mechanistic-clinical covariance explained. The outer sector contains the specific mechanisms, while the middle sector is grouped by receptor families and the inner sector by target biological factor.

neuroimaging-only models (see Methods: Regional influence). At each brain region, we studentized residuals across all patients, with each residual representing the unexplained pathology in a region at a given imaging visit. Then, for all regions, we computed the Wilcoxon rank-sum statistics of the population residuals from the two models, and repeated the model-fitting procedure with 1000 randomly shuffled receptor maps to obtain a null distribution of Wilcoxon statistics. We used this permutation test to filter brain regions with significant residual improvements ($P < 0.05$) over the null distributions. These maps do not represent the regions with the highest pathological severity, but rather those where longitudinal alterations are significantly better explained by the inclusion of a particular receptor distribution. In Fig. 7, we summarize the receptor influence maps for the top 4 receptor-pathology pathways (Fig. 3a): $5HT_2$ and $M_2$ on microstructural alterations (MD), $\alpha_4\beta_2$ on gray matter atrophy (GM), and kainate on dendrite density (t1/t2). Receptor influence maps for all biological factors are presented in Supplementary Figs. S1–S6.

Among other regions, the $5HT_2$ receptor most prominently influences microstructure (MD) in the anterior and medial thalamus, left posterior cingulate region (Brodmann area 31), anterior prefrontal cortex, left primarily motor cortex, right premotor cortex and supplementary motor area (Brodmann area 6). The muscarinic $M_2$ receptor influences microstructural alterations in the somatosensory cortex, left distal visual area V3d, right primary motor cortex, left hippocampus (CA), right primary somatosensory cortex (Brodmann area 2), lateral prefrontal cortex (Brodmann areas 46 - left and 47 - right), and entorhinal cortex (Brodmann areas 36-right and 37-left). The nicotinic $\alpha_4\beta_2$ receptor influences gray matter atrophy in the (left and right) thalamus, primary somatosensory cortex (Brodmann area 2), right temporal inferior parietal area, left caudate nucleus and entorhinal cortex (left Brodmann region 22). Kainate influences dendrite density in a broad set of regions, focused on the thalamus, visual areas (V1, V2 and the ventral parts of V3 and V4 in the right hemisphere, and V1 and ventral V4 in the left hemisphere), and prefrontal areas.

Across biological factors, glutamatergic receptors contribute significantly to explaining neurodegeneration in fronto-temporal regions (Supplementary Fig. S1). Particularly, both AMPA and kainate receptors contribute strongly to most factors (except for dopamine transporter loss) in frontal regions. The influences of $GABA_A$ receptors, $GABA_B$ receptors and the benzodiazepine binding site (Bz site) generally follow their distribution (Supplementary Fig. S2), peaking at visual, visual-parietal and fronto-temporal areas, respectively. Notably, dendrite loss is most pronounced at subcortical and fronto-temporal regions for all GABAeric receptors.

## Discussion

The complex pathophysiology of PD involves multiple difficult-to-map neurotransmitter systems, and the selective vulnerability of various non-dopaminergic nuclei[4]. We apply a personalized, causal brain modeling approach that combines longitudinal neuroimaging data and clinical assessments with averaged spatial receptor templates, to infer the previously uncharacterized roles of receptor-mediated interactions in PD neurodegeneration and symptomatic heterogeneity.

In PD, dopaminergic neuroimaging is common[36], and some non-dopaminergic targets such as acetylcholinesterase have been characterized[37]. However, the expense of PET imaging and the lack of suitable in vivo radioligands have impeded the study of many other receptor alterations in a PD population. Our method circumvents this limitation by inferring the importance of receptor interactions in individualized models of brain reorganization. We note that the different receptor maps are not very correlated with each other (Supplementary Figs. S1–S6) and the "multi-receptor fingerprint" of each (cyto-architectonically defined) brain region is distinct, particularly differing across the functional hierarchy[31]. In vitro multi-receptor autoradiography of the caudate nucleus and midcingulate area 24 of progressive supranuclear palsy (PSP) patients showed differentiation of patients from age-matched controls, as well as diverging alterations

**a  Receptor densities**

5HT$_2$ | M$_2$ | α$_4$β$_2$ | Kainate

**b  Receptor influence on biological factors**

5HT$_2$ → Undirected microstructure | M$_2$ → Undirected microstructure | α$_4$β$_2$ → Gray matter density | Kainate → Dendrite density

**Fig. 7 | Model-derived maps of receptor influence on PD neurodegeneration.**
We compared the (**a**) receptor densities and (**b**) influence maps. Influence maps show the brain regions where specific receptors are consistently informative to explaining the neuropathological changes across all PD subjects, and are re-scaled to arbitrary units for visualization. They represent the population-wide improvement in model residuals at each region due to the inclusion of receptor density maps and receptor-pathology interactions as model predictors for each PD patient. Receptor influences are calculated as the Wilcoxon rank-sum statistics of each model's residuals for a region, and the maps show only regions with significant $z$-scores ($P < 0.05$) of Wilcoxon rank-sum statistics relative to the null distributions.

in clinical subgroups of PSP[38]. In this related movement disorder, notable, previously unknown receptor associations (to kainate and adenosine type 1 receptors) were discovered, supporting the case for more thorough receptor mapping studies in neurodegenerative populations.

Lacking in vivo or in vitro receptor mapping data in PD patients, we attempted to use in silico modeling to infer regional susceptibility to neurodegeneration based on receptor expression, and characterize the relationship between inter-individual variability in receptor-mediated neurodegeneration and symptomatic variability. Recent works in Alzheimer's disease (AD) have demonstrated that model parameters from personalized brain models can represent otherwise unobservable, latent mechanisms that relate to phenotype better than raw imaging data[32,39,40].

While we used autoradiography-derived templates of receptor density, receptor gene expression may be used as a proxy[41]. For example, the Allen Human Brain Atlas (http://human.brain-map.org) gene expression template has been used to identify transcriptomic pathways mediating neurodegeneration in AD[40]. However, several translational and trafficking steps separate gene expression and synaptically integrated receptors. Although receptor densities and gene expression are correlated for selected receptor subunit genes and across certain cytoarchitectonically-defined regions[42], this is not universally true[43]. Low correlations are also observed between gene expression and in vivo PET imaging of dopamine transporters[44]. Other works combining unimodal neuroimaging from disease cohorts with PET- and SPECT-derived healthy neurotransmitter receptor and transporter templates have uncovered the co-localization of specific neurotransmitter systems with PD resting state fMRI alterations[45], dyskinesia- and parkinsonism-associated atrophy in schizophrenia patients[46], gray matter atrophy in symptomatic FTD and its genetic subtypes[47], and functional alterations in behavioral variant FTD[48]. Furthermore, our averaged autoradiography-derived receptor templates are correlated with neurobiological processes such as drug-induced cerebral blood flow changes[49]. In addition, in vitro autoradiography allows access to a broader class of receptors (without in vivo ligands) at a sub-millimeter resolution (as low as 0.3 mm slice width per receptor[31]) compared to PET with its theoretical bound of ~2 mm spatial resolution[50]. Future work will extend the presented

results with voxel-scale whole brain receptor maps rather than macroscopically averaged values.

We incorporated several neuroimaging-derived measures sensitive to PD progression[51], from structural MRI-based gray matter density (GM) and dendrite density (t1/t2 ratio), diffusion-based measures of microstructural integrity (MD and FA)[52], functional neuronal activity (fALFF) and presynaptic dopamine transporter availability (DAT-SPECT). Resting-state fMRI-derived metrics such as fALFF can distinguish PD patients from controls[53], with fALFF being able to explain up to 25% of variability in MDS-UPDRS scores[54]. While initially proposed as a quantitative measure of demyelination from routine MRI scans, t1/t2 ratio has since been demonstrated to have a stronger correlation with dendritic density[33,34], particularly relevant to synaptic integrity and receptor activity. Furthermore, our flexible modeling approach can be extended to incorporate other relevant modalities.

Although receptor maps were averaged from neurologically healthy aged brains, earlier work has demonstrated their utility in other cohorts, namely healthy aged subjects, mildly cognitively impaired subjects, and AD patients from the Alzheimer's disease Neuroimaging Initiative (ADNI)[32]. Extending this validation to the PPMI cohort, we note an ~42.3% improvement in the explanation of neuro-pathology accumulation in receptor-enriched models. These improvements are statistically significant for well over 90% of subjects ($P < 0.05$ in both $F$-tests and permutation tests; Fig. 2c, d) for all biological factors with the exception of undirected microstructural damage (MD). We used a non-parametric permutation to generate null receptor distributions, which does not consider any spatial auto-correlation in the receptor maps[55]. The lack of voxel-scale receptor maps and the inclusion of subcortical regions in our parcellation would preclude both cortical surface rotation-based methods as well as parametrized models requiring autocorrelation information in other, more typical statistical analyses.

For a third of all subjects, the improvement in model fit of undirected microstructure was not significantly better than permuted null distributions of receptors. While diffusion MRI can be sensitive to aspects of gray matter microstructure[56,57], it is less accurate than in white matter due to the heterogeneity of tissues and their (lack of) organization[58]. Yet, despite the limitation of partial volume effects in gray matter ROIs[59], receptor-enriched models fit longitudinal

alterations to microstructure reasonably well (average $r^2 = 0.70$ for undirected MD and $r^2 = 0.74$ for directed FA; Fig. 2).

Differential neurotransmitter and receptor expression may underpin the selective vulnerability of several neuronal populations, from the dopaminergic substantia nigra to the adrenergic locus coeruleus and serotonergic raphe nuclei, and their cortical projections[60]. Furthermore, PD neurodegeneration may alter both the spatial distributions as well as functional interactions of specific dopaminergic and non-dopaminergic receptors, with symptomatic consequences[14]. In our mechanistic modeling framework, each model weight is interpretable as the importance of specific neurobiological mechanism. Receptors contribute to neurodegeneration in re-MCM either as (i) direct effects representing regional susceptibility to neurodegeneration based on receptor expression, or (ii) receptor-mediated interactions involving a source and target biological factor. In addition, biological factors have (i) local effects on themselves and other factors, and (ii) intra-factor network effects due to propagation of pathology along the structural connectome. Lacking inter-subject variability in receptor data, our model compensates by assigning weights differently across subjects. Consistent trends in model weights reflect the importance of the corresponding neurobiological mechanism across the PD population, while co-variability with symptoms suggests clinical relevance.

First, we identified specific mechanisms affecting neurodegeneration across the PD cohort (Fig. 3). We observed a complex network of interactions between biological factors, with distinct receptor profiles affecting each factor. The large contributions of receptor-mediated inter-factor interactions (Fig. 3a) supports the multi-system view of PD. Fewer receptors are statistically stable predictors of longitudinal changes to functional activity (fALFF), directed microstructural damage (FA) and dopaminergic neurotransmission (SPECT), while gray matter atrophy (GM), dendrite density (t1/t2 ratio) and undirected microstructural changes (MD) show greater influence from a more diverse set of receptors.

Notably, the $D_1$ receptor map is not a stable predictor of DAT alterations. While presynaptic DAT density and postsynaptic dopaminergic receptor distributions are strongly related under normal conditions, they may be affected differently by disorders. For example, while $D_2$ receptor availability is reduced in alcoholism, DAT availability is preserved[61]. In PD, DAT-SPECT and receptor PET imaging have distinct clinical interpretations[62], and increased dopamine turnover early at symptom onset has implicated presynaptic mechanisms at this disease stage[63]. Furthermore, healthy aged $D_1$ receptor expression is relatively uninformative as it is comparatively homogeneous across cortical regions (Supplementary Fig. S6) and likely redundant to the model in the presence of individualized DAT imaging. On the other hand, DAT density also peaks in striatal regions, and DAT-SPECT is not able to resolve cortical radiotracer uptake as well as DAT-PET[64]. SPECT is currently more prevalent clinically for DAT imaging, and was thus the modality used in a large, multi-center study such as PPMI. Nevertheless, it must be noted that DAT-SPECT is limited in its ability to resolve cortical alterations, and this is likely reflected in its under-emphasis in our results.

Network degeneration hypotheses of PD pathogenesis implicate various mechanisms from the propagation of neurotoxic alpha-synuclein[65] to the structural and functional neurodegeneration following striatal denervation[66]. We note that propagation is only a small contributor to the accumulation of pathology, and is dwarfed by local effects in our models (Fig. 3a). These findings may potentially reflect distinct disease phases. Our cohort was composed entirely of PD patients, for whom propagative, disease seeding processes may have already occurred, and neurodegeneration may now be driven by local effects. Furthermore, white matter tractography may not completely capture the connectivity between our cyto- and receptor-architectonically defined regions. A more complete treatment may consider vascular connectivity as well[29,30], which may also be a substrate for pathology propagation.

We find notable glutamatergic effects on multiple (micro-)structural factors (Supplementary Fig. S1): gray matter atrophy (NMDA), directed microstructural damage (AMPA and kainate), and dendrite density (kainate and NMDA). As NMDA and AMPA receptors are postsynaptic targets of glutamate, these mechanisms likely reflect the structural consequences of excitotoxicity and cell death[67]. On the other hand, kainate is believed to modulate synaptic transmission and plasticity[68], which may affect dendritic density. In our models, NMDA receptor influence is focused on occipital and temporal regions, AMPA influence is highest in frontal regions, and kainate influences mainly dendrite loss in both frontal and occipital regions. Among glutamatergic receptors, influence on microstructure of the motor cortex (MD, FA and t1/t2) is prominent, although it is more limited for atrophy or functional alterations.

The stable roles of GABAergic receptors (Fig. 3a) suggest their involvement via altered neuronal activity inhibition, interaction with the dopaminergic system, and potential regional vulnerability to microstructural degradation or dendrite loss. Inter-subject variability along the primary, mainly motor axis correlates with GABAergic mechanisms affecting microstructure (FA, MD and t1/t2) and functional activity. Furthermore, a magnetic resonance spectroscopy (MRS) study found reduced levels of GABA in the visual cortex of PD patients[69], consistent with the regions of maximal influence of $GABA_A$ and $GABA_B$ receptors in our model.

Due to the necessity for sufficient longitudinal and multi-modal scans, no healthy subjects met our inclusion criteria. As each individualized model is fit independently, we account for the confounding effects of healthy ageing on model-derived mechanisms by performing a multivariate correlation with 11 assessments representing various symptomatic domains, with age as a covariate. Presently, PD is defined primarily by clinical symptoms, and thus any combination of model mechanisms robustly correlated with multi-domain symptoms can be considered as contributing to the spectrum of PD rather than healthy (i.e., non-symptomatic) aging.

Various non-dopaminergic neurotransmitter systems have been associated with specific symptoms in PD, including cholinergic memory defects, adrenergic impairment of attention, and serotonin-driven depression[70] and visual hallucinations[71,72]. Comparing model-derived receptor mechanisms and clinical assessments across PD patients, we observe two main axes of co-variability. The primary component represents motor/psychomotor symptoms associated prominently with GABAergic mechanisms, with secondary contributions from glutamatergic, cholinergic, and serotonergic systems (Supplementary Table S7). The secondary component is defined by visuospatial, memory and psychiatric symptoms, with the cholinergic system being the dominant receptor family. Mechanisms affecting microstructure (FA and MD) are more prominent in the primary component, while those affecting gray matter density are greater in the secondary component. Nevertheless, receptor mechanisms affecting microstructure and dendrite density (t1/t2) contribute strongly to both axes.

The secondary component is consistent with the cholinergically-driven visuospatial aspect of the dual-syndrome hypothesis of PD[18]. Stable cholinergic mechanisms are also present for every biological factor except directed microstructure (FA), most notably the contributions to dendrite loss (t1/t2), undirected microstructural damage (MD) and gray matter atrophy (Fig. 3a). Specifically, we note prominent muscarinic $M_2$ and nicotinic $\alpha_4\beta_2$ receptor influences (on MD and GM, respectively) on the primary somatosensory cortex, a site of reduced activation in PD (Fig. 7)[73]. Our model suggests that nicotinic and muscarinic cholinergic systems strongly affect PD symptoms along specific pathways primarily involving dendritic density, atrophy, and degradation of microstructure (Fig. 6b). While typically associated

with cognitive impairment and dementia in PD, cholinergic degeneration is also linked to depressive mood, apathy, olfaction, sleep disorder, and postural and gait disorder[74]. Epidemiological studies of smokers suggest a neuroprotective role for nicotinic receptors[75], which experience widespread decrease in PD[76]. The cholinergic and dopaminergic systems interact at biochemical, circuit and functional levels[70], tightly coupled by nicotinic receptors expressed on striatal dopaminergic neurons and acetylcholine[70,77] modulate dopaminergic neurotransmission. An imbalance of cholinergic and dopaminergic neurotransmission may thus underlie PD cognitive dysfunction[70]. Our results suggest that cholinergic receptor distributions contribute to both motor and non-motor axes, albeit via distinct pathways (Fig. 6a,b).

We note the mild motor phenotype of the PD patients from PPMI included in this work (mean MDS-UPDRS Part III score, Supplementary Table S1). Potential low variability in these scores in combination with the poor cortical signal in DAT-SPECT may have under-emphasized the dopaminergic-motor axis of PD. Nevertheless, the dopaminergic relationship with motor symptoms is reproduced in the primary, mainly motor component, with DAT-SPECT appearing as a target imaging modality. In addition to the classical dopaminergic-motor axis, our work presents a multi-modal perspective of PD, associating multivariate combinations of receptor distributions with macroscopic imaging-derived pathological alterations, and motor and non-motor symptoms.

In addition to mediating inter-factor interactions, dysfunctional interactions between receptors may also be involved in neurodegeneration. Neurotransmitter release is regulated by presynaptic auto- and hetero-receptors[78], which in PD is potentially impaired in the dopaminergic system[79] and in GABAergic inhibition of the motor cortex[80]. Where possible, concurrent receptor or transporter imaging in a PD cohort would help clarify the role of neurotransmission balance in neurodegeneration.

We attempted to cover a broad variety of (particularly structural) disease-sensitive neuroimaging modalities. Yet, PD neurodegeneration is complex and likely also involves changes to surface morphology[81,82], such as gyrification. However, to include the basal ganglia and thalamus in our model using the same set of features, we did not include surface-based measures.

Despite the prevalence of PD, the causes of this neurodegenerative condition remain unknown, and treatment is limited to symptomatic therapy complicated by individual variability in clinical presentation, side effects and treatment response[83]. Our work sheds light on the complex, especially non-dopaminergic neurotransmitter receptor-mediated mechanisms underlying brain reorganization and symptomatic variability in PD. As longitudinal data collection progresses in large cohorts, model-derived mechanisms may help differentiate mechanisms distinct to PD and its (genetic or clinical) subtypes, Parkinson-plus syndromes, other neurodegenerative diseases, and healthy ageing. Since neurotransmitter receptors are clinically efficacious drug targets[8], future work will explore the use of our personalized modeling approach to design personalized receptor-based therapy.

## Methods
### Ethics statement
This work has been conducted in accordance with ethical guidelines and regulations. Neuroimaging and clinical data in this study was acquired through the multi-center Parkinson's Progression Markers Initiative (PPMI; ppmi-info.org). Following good clinical practices and in accordance with the Declaration of Helsinki guidelines, study subjects and/or authorized representatives gave written informed consent at the time of enrollment for sample collection and completed questionnaires approved by each participating site Institutional Review Board (IRB). The authors obtained approval from the PPMI for data use

and publication, see documents https://www.ppmi-info.org/documents/ppmi-data-use-agreement.pdf and https://www.ppmi-info.org/documents/ppmi-publication-policy.pdf, respectively.

### Data description and processing
**Study participants.** This study used longitudinal data from $N = 71$ participants from the PPMI from 12 international sites, with a clinical diagnosis of PD. Demographic information is summarized in Supplementary Table S1. The inclusion criterion was the presence of at least three different imaging modalities (i.e., structural MRI, resting functional MRI, diffusion MRI and/or dopamine SPECT) over at least three visits at the time of our analysis.

**Structural MRI acquisition/processing.** Brain structural T1- and T2-weighted 3D images were acquired for all $N = 71$ subjects. A detailed description of acquisition details can be found from the PPMI procedures manuals at http://www.ppmi-info.org/. T1- and T2-weighted images from 3 T scanners were acquired as a 3D sequence with a slice thickness of 1.5 mm or less, under three different views: axial, sagittal and coronal. All images underwent non-uniformity correction using the N3 algorithm[84]. Next, they were segmented into gray matter probabilistic maps using SPM12 (version 12, https://fil.ion.ucl.ac.uk/spm). Gray matter segmentations were standardized to MNI space[85] using the DARTEL tool[86]. Each map was modulated to preserve the total amount of signal/tissue. Mean gray matter density[86] values were calculated for the regions described in Methods: Data description and processing: Receptor densities and brain parcellation.

**Resting fMRI acquisition/processing.** Resting-state functional images were obtained using an echo-planar imaging sequence on 3 T MRI scanners for $N = 71$ subjects. For a detailed description of acquisition protocols, please see http://www.ppmi-info.org. Acquisition parameters were: 140 time points, repetition time (TR) = 2400 ms, echo time (TE) = 25 ms, flip angle = 80°, number of slices = 40, slice thickness = 3.3 mm, in plane resolution = 3.3 mm and in plane matrix = 68 × 66. Pre-processing steps included: (1) motion correction, (2) slice timing correction, (3) alignment to the structural T1 image, and (4) spatial normalization to MNI space using the registration parameters obtained for the structural T1 image with the nearest acquisition date, and (5) signal filtering to keep only low frequency fluctuations (0.01–0.08 Hz)[87]. For each brain region, our model requires a local (i.e., intra-regional, non-network) measure of functional activity, to maintain mechanistic interpretability and to prevent data leakage of network information into local model terms (described further in Receptor-Enriched Multifactorial Causal Model). Due to its high correlation with glucose metabolism[88] and disease progression in PD[53], we calculated regional fractional amplitude of low-frequency fluctuation (fALFF)[89] as a measure of functional integrity.

**Diffusion MRI acquisition/processing.** Diffusion MRI (dMRI) images were acquired using standardized protocol on 3 T MRI machines from 32 different international sites. Diffusion-weighted images were acquired along 64 uniformly distributed directions using a b-value of 1000 s/mm² and a single b = 0 image. Single shot echo-planar imaging (EPI) sequence was used (116 × 116 matrix, 2 mm isotropic resolution, TR/TE 900/88 ms, and twofold acceleration). An anatomical T1-weighted 1 mm³ MPRAGE image was also acquired. Each patient underwent two baseline acquisitions and a further two 1 year later. More information on the dMRI acquisition and processing can be found online at http://www.ppmi-info.org/. Preprocessing steps included: (1) motion and eddy current correction[90], (2) EPI distortion correction, (2) alignment of the T1-weighted image to the b0 image based on mutual information, (3) calculation of the deformation field between the diffusion and T1-weighted images, (4) calculation of the voxelwise diffusion tensors, (5) alignment to the structural T1 image,

and (6) spatial normalization to MNI space[85] using the registration parameters obtained for the structural T1 image with the nearest acquisition date, and (6) calculation of mean values of summary metrics (FA and MD) for each considered brain region.

**Dopamine SPECT acquisition/processing.** A 111–185 MBq (3–5 mCi) bolus injection of I-123 FB-CIT was administered to each participant ($N = 71$), and the SPECT scan was performed 4 h post-injection. Raw projection data was acquired as a $128 \times 128$ matrix and the SPECT image was reconstructed. Attenuation correction and Gaussian blurring with a 3D 6 mm filter were applied. The reconstructed and corrected SPECT images were normalized and registered to MNI space[85], and average values were calculated for all considered regions of interest.

**Receptor densities and brain parcellation.** In vitro quantitative receptor autoradiography was applied to measure the densities of 15 receptors in 57 cytoarchitectonically defined cortical areas spread throughout the brain[91]. These receptors span major neurotransmitter systems and show significant regional variability across the brain. Brains were obtained through the body donor programme of the University of Düsseldorf. Donors (three male and one female; between 67 and 77 years of age) had no history of neurological or psychiatric diseases, or long-term drug treatments. Causes of death were non-neurological in each case. Each hemisphere was sliced into 3 cm slabs, shock frozen at −40C, and stored at −80C.

Receptors for the neurotransmitters glutamate (AMPA, NMDA, kainate), GABA (GABA$_A$ GABA$_A$-associated benzodiazepine binding sites, GABA$_B$), acetylcholine (muscarinic M$_1$, M$_2$, M$_3$, nicotinic $\alpha_4\beta_2$), noradrenaline ($\alpha_1$, $\alpha_2$), serotonin (5-HT$_{1A}$, 5-HT$_2$), and dopamine (D$_1$) were labeled according to previously published binding protocols consisting of pre-incubation, main incubation and rinsing steps[91]. The ligands used are summarized in Supplementary Table S3. Receptor densities were quantified by densitometric analysis of the ensuing autoradiographs, and areas were identified by cytoarchitectonic analysis in sections neigbouring those processed for receptor autoradiography, and which had been used for the visualization of cell bodies[92].

A brain parcellation was then defined with the aid of the Anatomy Toolbox[93] using 57 regions of interest for which receptor densities were available[31]. This parcellation was based on areas identified by cortical cytoarchitecture, as well as other cyto- and receptor-architectonically defined regions with receptor measurements (regions are summarized in Supplementary Table S4). These 57 regions were mirrored across left and right hemispheres for a total of 114 brain regions in our parcellation. For each receptor, regional densities were normalized using the mean and standard deviation across all brain regions.

The structural T1 images of the Jülich[93], Brodmann[94], AAL3[95] and DISTAL[96] brain parcellations were registered to the MNI ICBM152 T1 template using the FSL (version 6.0) FLIRT affine registration tool[97], and the obtained transformations were used to project the corresponding parcellations to the MNI ICBM152 space (using nearest neighbor interpolation to conserve original parcellation values). In the MNI ICBM152 space, voxels corresponding to the cytoarchitectonically-defined regions from[31] were identified from the regions in the Anatomy Toolbox, with the remaining Brodmann regions filled in using the Brodmann brain atlas. Supplementary Table S4 summarizes the ROI maps used to create the Brain atlas for regions with receptor data. The resulting parcellation of 114 brain regions in the common template space was then quality controlled, and small regions under 50 voxels were excluded. The resulting atlas with 155 bilateral brain regions (95 of which had receptor data) was used to extract whole-brain multi-modal neuroimaging data and estimate the diffusion-based connectivity matrix, as described in

Methods: Multimodal neuroimaging data fusion and Methods: Anatomical connectivity estimation.

**Anatomical connectivity estimation.** The connectivity matrix was constructed using DSI Studio (March 8, 2019 build; http://dsi-studio.labsolver.org). A group average template was constructed from a total of 1065 subjects[98]. A multi-shell diffusion scheme was used, and the b-values were 990, 1985 and 2980 s/mm$^2$. The number of diffusion sampling directions were 90, 90, and 90, respectively. The in plane resolution was 1.25 mm. The slice thickness was 1.25 mm. The diffusion data were reconstructed in the MNI space using q-space diffeomorphic reconstruction[99] to obtain the spin distribution function[100]. A diffusion sampling length ratio of 2.5 was used, and the output resolution was 1 mm. The restricted diffusion was quantified using restricted diffusion imaging[101]. A deterministic fiber tracking algorithm[102] was used. A seeding region was placed at whole brain. The QA threshold was 0.159581. The angular threshold was randomly selected from 15 degrees to 90 degrees. The step size was randomly selected from 0.5 voxel to 1.5 voxels. The fiber trajectories were smoothed by averaging the propagation direction with a percentage of the previous direction. The percentage was randomly selected from 0% to 95%. Tracks with length shorter than 30 or longer than 300 mm were discarded. A total of 100,000 tracts were calculated. A custom brain atlas based on cytoarchitectonic regions with neurotransmitter receptor data[31] was used as the brain parcellation, as described in Methods: Data description and processing: Receptor densities and brain parcellation, and the connectivity matrix was calculated by using count of the connecting tracks.

**Multimodal neuroimaging data fusion.** After pre-processing PPMI neuroimaging data for all 6 modalities, data harmonization was performed using ComBat (commit 91f8bf3, https://github.com/Jfortin1/ComBatHarmonization)[103]. Each site used the same scanner for all subjects, and our harmonization procedure corrected for batch effects due to sites while preserving variance due to clinical diagnosis, age, education level, sex and (left or right) handedness. After extracting harmonized neuroimaging data for the cytoarchitectonically defined atlas described in Methods: Data description and processing: Receptor densities and brain parcellation, subjects lacking sufficient longitudinal or multimodal data were discarded. The disqualification criteria were (i) fewer than 4 imaging modalities with data, or (ii) fewer than 3 longitudinal samples for all modalities. For the remaining subjects, missing neuroimaging modalities (primarily FA, MD and t1/t2 ratios) at each visit were imputed using trimmed scores regression. Finally, a total of $N = 71$ subjects were left with all 6 neuroimaging modalities with an average of 3.59 ($\pm 0.50$) time points. We used the mean and variance of each neuroimaging modality across all regions to calculate z-scores of neuroimaging data for all subjects. Please see Supplementary Table S1 for demographic characteristics.

**Clinical scores.** We used multiple composite scores derived from the PPMI clinical (motor, non-motor, psychiatric, cognitive, etc.) testing battery, namely the Benton Judgment of Line Orientation Test (BJLOT[104]), Geriatric Depression Scale (GDS[105]), Hopkins Verbal Learning Test (HVLT[106]), Letter Number Sequencing (LNS[107]), Movement Disorders Society – Unified Parkinson's Disease Rating Scale (MDS-UPDRS[108]) Parts 1 (non-motor aspects of daily living; NP1), 2 (motor aspects of daily living; NP2), and 3 (motor exam; NP3), the Montreal Cognitive Assessment (MoCA[109]), semantic fluency (SF), State-Trait Anxiety Inventory for Adults (STAIAD[110]), and Symbol Digit Modalities (SDM[111]) tests. Protocols for deriving each score are described in the respective PPMI protocols documentation. We calculated symptomatic decline as the rate of change (linear slope) of the 11 clinical scores with respect to examination date. Average numbers of

longitudinal evaluations per clinical score are summarized in Supplementary Table S2.

**Receptor-enriched multifactorial causal model (re-MCM).** Multifactorial causal modeling is a generalized framework[29,32] that treats the brain as a dynamical system of ROIs characterized by multiple interacting neuroimaging-quantified biological factors. Pathology may develop over time in each factor, affecting other factors locally and propagating to neighbouring regions via anatomical connections. We introduce the receptor-enriched multifactorial causal model (re-MCM), in which the local densities of various neurotransmitter receptors mediate interactions between biological factors at each brain region.

In this work, the biological factors are gray matter density, neuronal activity, presynaptic dopamine, demyelination/dendritic density and two measures of white matter integrity, derived from structural T1 MRI, resting state functional MRI (rs-fMRI), DAT-SPECT, T1/T2 ratio, FA and MD, respectively. For any given subject and at a particular brain region $i$, the level of pathology of each biological factor $m$ is represented by a single variable $S_{m,i}$, calculated as the deviation from the neuroimaging signal at the baseline visit. The temporal evolution of pathology $S_{m,i}$ in modality $m$ at brain region $i$ is given by following differential equation:

$$\frac{dS_{m,i}(t)}{dt} = \underbrace{f(\mathbf{S}_{*,i}(t), \mathbf{R_i})}_{\text{Local Effects}} + \underbrace{g(\mathbf{S}_{m,*}(t), \mathbf{C}_{i\leftrightarrow*})}_{\text{Inter-region Propagation}} \quad (1)$$

The functions $f$ and $g$ govern the global biological factor dynamics that are consistent across all brain regions. The local component $f(S_{,i}(t), R_{,i})$ is the cumulative effect of all biological factors on factor $m$ within region $i$ mediated by $\mathbf{R}_i$, composed of local densities $r_{k,i}$ of a receptor $k$ at a region $i$. The propagation term $g$ represents the net spreading of pathology in factor $m$ along anatomical connections $C_{i\leftrightarrow*}$ of the region $i$. Since the inter-visit interval of ~6 months is significantly shorter than the temporal scale of neurodegeneration, we assume a locally linear, time-invariant dynamical system:

$$\frac{dS_i^m(t)}{dt} = \sum_{n=1}^{N_{\text{fac}}} \alpha^{n\to m} S_{n,i}(t) + \sum_{k=1}^{N_{\text{rec}}} \alpha_k^m r_{k,i}$$
$$+ \alpha_{\text{prop}}^m \sum_{j=1, j\neq i}^{N_{\text{ROI}}} \left[ C_{j\to i} S_{m,j}(t) - C_{i\to j} S_{m,i}(t) \right], \quad (2)$$

where $C_{i\to j}$ is the directed anatomical connectivity from region $i$ to $j$, and $\frac{dS_{m,i}(t)}{dt}$ the local rate of change of neuroimaging data for successive longitudinal samples at times $t'$ and $t$:

$$\frac{dS_{m,i}(t)}{dt} = \frac{S_{m,i}(t) - S_{m,i}(t')}{t - t'} \quad (3)$$

Local effects include (i) direct factor-factor effects, (ii) interaction terms mediated by $N_{\text{rec}} = 15$ receptor types, and (iii) direct receptor effects on the biological factor rate of change $\frac{dS_{m,i}}{dt}$ (the second term in Eq. 2). The first term in Eq. 2 is expanded as:

$$\alpha^{n\to m} = \underbrace{\alpha_0^{n\to m}}_{\text{Direct Factor-Factor Term}} + \underbrace{\sum_k^{N_{\text{rec}}} \alpha_k^{n\to m} r_i^k}_{\text{Interaction Term}} \quad (4)$$

The propagation term assumes symmetric connectivity $C_{j\leftrightarrow i}$ between regions $i$ and $j$, using a template connectivity matrix for all subjects, as described in Anatomical connectivity estimation, so we

define the propagation component as:

$$p_{m,i}(t) = \sum_{j=1, j\neq i}^{N_{\text{ROI}}} C_{j\leftrightarrow i} \left[ S_{m,j}(t) - S_{m,i}(t) \right] \quad (5)$$

Thus, for each subject, the evolution of pathology in each biological factor $m$ at region $i$ is described by:

$$\frac{dS_{m,i}(t)}{dt} = \sum_{n=1}^{N_{\text{fac}}} \left( \alpha_0^{n\to m} + \sum_k^{N_{\text{rec}}} \alpha_k^{n\to m} r_{k,j} \right) S_{n,i}(t)$$
$$+ \sum_{k=1}^{N_{\text{rec}}} \alpha_k^m r_{k,i} + \alpha_{\text{prop}}^m p_{m,i}(t) \quad (6)$$

Each model contains a set of $N_{\text{params}} = N_{\text{fac}} \times (1 + N_{\text{rec}}) + N_{\text{rec}} + 1 = 113$ parameters $\{\alpha\}_x^m$ for subject $x$ and factor $m$ (or 678 total parameters per subject), each with a distinct neurobiological interpretation (e.g., the effect of reduced white matter integrity on gray matter atrophy mediated by glutamatergic receptor density). We perform linear regression, using the terms in Eq. 6 as predictors with longitudinal PPMI neuroimaging samples $S_{m,i}(t)$ and receptor maps $R$, to fit parameters $\{\alpha\}_x^m$ for each subject $x$ and modality $m$. Separate regression models were built for (i) each of the $N = 71$ qualifying subjects, and (ii) each of the 6 neuroimaging factors. These subjects were drawn from the PPMI dataset with at least 3 recorded neuroimaging modalities, and at least 3 longitudinal samples for at least one modality.

We then calculate the coefficient of determination ($R^2$ for each model to evaluate model fit, summarized in Fig. 2. With the true neuroimaging-derived data $y_{m,i,t} = \frac{dS_{m,i}(t)}{dt}$, subject-wise mean imaging data $<y_m>$ for modality $m$ across all brain regions and longitudinal samples, and model predictions $\hat{y}_{m,i,t}$, the coefficient of determination is

$$R^2 = 1 - \frac{\sum_{it}(y_{m,i,t} - \hat{y}_{m,i,t})^2}{\sum_{it}(y_{m,i,t} - <y_m>)^2} \quad (7)$$

**Model fit.** For each subject and neuroimaging modality, we evaluated the quality of model fit by calculating the coefficient of determination ($R^2$). Secondly, to evaluate the improvement in model fit due to receptor and receptor-mediated interaction terms while accounting for the difference in model size for each subject, we used $F$-tests ($p < 0.05$) to compare the model fit of the full, receptor-neuroimaging interaction models (113 parameters per modality) with restricted, neuroimaging-only (8 parameters per modality) models.

Finally, we evaluated the significance of the improvement in model fit ($R^2$) due to actual receptor distributions with a permutation test using 1000 iterations of randomly permuted receptor maps (with receptor densities shuffled across regions independently for each receptor type), calculating the $p$ value of the model $R^2$ with the true receptor data compared to the null distribution.

**Covariance of biological mechanisms with clinical symptoms.** To identify multivariate links between receptor-mediated biological mechanisms and to clinical symptoms in PD, we performed a data-driven partial least squares (PLS) regression analysis. Using singular value decomposition (SVD) to factorize the population covariance matrix between re-MCM parameters and clinical assessments (summarized in Methods: Clinical scores) to its eigenvectors, we identify multivariate axes of co-varying features. Different axes represent orthogonal disease processes affecting symptom severity. Permutation tests and bootstrapping ensure the statistical significance of the axes and the stability of identified mechanisms and symptoms, respectively. The algorithm is summarized as follows:

1. We performed SVD on the cross-covariance matrix between all 678 re-MCM parameters and rates of clinical decline for $N = 71$ PD patients, adjusted for covariates (baseline age, education, and sex). SVD simultaneously reduces the dimensionality of features, and ranks them by their contribution to each axis. The cross-covariance matrix $C = XY'$ of the $z$-scores of re-MCM parameters $X$ and the $z$-scores of the clinical decline rates $Y$ is decomposed as

$$C = USV' \tag{8}$$

where $U$ and $V$ are orthonormal matrices of spatial loadings for the parameters and clinical scores, respectively, and $S$ is a diagonal matrix of singular values $\{s_1, \ldots, s_7\}$.

2. We then performed permutation tests by shuffling the mapping between subjects' re-MCM parameters and clinical scores, and repeating Step 1 for 1000 iterations, to evaluate the significance of latent components. We performed a Procrustes transformation to align the axes of singular components in order to compare components from permuted iterations. We retained only those significant ($p < 0.05$ with respect to the permuted distribution) singular components.

3. To discard non-stable re-MCM parameters and clinical assessments in each axis, we performed 1000 iterations of bootstrapping on the parameters $X$ and clinical scores $Y$. To compare permuted iterations, we performed a Procrustes transformation to align the axes of singular components. We discarded the parameters with non-stable 95% confidence intervals.

4. For the remaining stable re-MCM parameters and clinical scores, and significant latent components, we computed the variance explained per parameter $j$ along each axis $i$:

$$r_{i,j}^2 = \underbrace{\frac{U_{i,j}^2}{\sum_j U_{i,j}^2}}_{\text{Parameter contribution}} \tag{9}$$

**Regional influence.** To infer the spatial patterns of receptor involvement in neurodegeneration, we examined the improvement in neuroimaging models due to the inclusion of each receptor map. For each biological factor $m$, receptor $k$ and brain region $i$, we fit a restricted, single-receptor version of the model

$$\frac{dS_{i,k}^m(t)}{dt} = \sum_{n=1}^{N_{fac}} \left( \alpha_0^{n \to m} + \alpha_k^{n \to m} r_{k,j} \right) S_{n,i}(t) + \alpha_k^m r_{k,i} \\ + \alpha_{prop}^m \sum_{j=1, j \neq i}^{N_{ROI}} \left[ C_{j \to i} S_{m,j}(t) - C_{i \to j} S_{m,i}(t) \right], \tag{10}$$

where the longitudinal rate of change of each factor is predicted by its network propagation, direct factor effects, the local density of a single receptor $k$, and factor interactions with the density of only receptor $k$. We compare this model with a restricted, neuroimaging-only model excluding receptor density and interactions:

$$\frac{dS_{i,k}^m(t)}{dt} = \sum_{n=1}^{N_{fac}} \alpha^{n \to m} S_{n,i}(t) + \alpha_{prop}^m \sum_{j=1, j \neq i}^{N_{ROI}} \left[ C_{j \to i} S_{m,j}(t) - C_{i \to j} S_{m,i}(t) \right] \tag{11}$$

To generate brain maps representing receptor influence on neuroimaging changes,

1. for each subject, we fit the single receptor and neuroimaging-only models for all biological factors and receptors, and studentize the residuals across regions and time points,

2. we combine the studentized residuals corresponding to each region across subjects and time points, and calculate the Wilcoxon rank-sum statistic $w_{i,k}^m$ between studentized residuals of the two models,

3. we compute a null distribution of the Wilcoxon statistic by repeating Steps 1–2 with 1000 randomly permuted receptor maps per imaging modality and receptor,

4. to estimate the significance of the Wilcoxon maps of each receptor across all 6 imaging modalities, we calculate the $z$-scores $z_{i,k}^m$ of the Wilcoxon statistic $w_{i,k}^m$ to its null distribution.

### Reporting summary

Further information on research design is available in the Nature Portfolio Reporting Summary linked to this article.

## Data availability

The three datasets used in this study are publicly available. The PPMI database (neuroimaging and clinical evaluations; https://www.ppmi-info.org/) is available to access after completing a data use agreement and submitting an online application (https://www.ppmi-info.org/access-data-specimens/download-data). The HCP database (HCP-1065[98]; tractography template for connectivity estimation; http://www.humanconnectomeproject.org/) is available at https://brain.labsolver.org/hcp_template.html, and receptor autoradiography data published in[31] is available at https://github.com/AlGoulas/receptor_principles. Source data are provided with this paper.

## Code availability

The PLS-SVD code is available at https://github.com/neuropm-lab/svd_pls. The re-MCM method (implemented in Matlab 2019b) will be incorporated as a part of our open-access, user-friendly software (https://www.neuropm-lab.com/neuropm-box.html)[112].

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

## Acknowledgements

The authors would like to acknowledge the integral role of the late Prof. Dr. Karl Zilles in collecting the receptor autoradiography data. This research was undertaken thanks in part to funding from: the Parkinson's Canada graduate training award to AFK, the *Canada First Research Excellence Fund*, awarded to McGill University for the *Healthy Brains for Healthy Lives Initiative*, the Canada Research Chair tier-2, *Fonds de la recherche en santé du Québec* (FRQS) Junior 1 Scholarship, Natural Sciences and Engineering Research Council of Canada (NSERC) Discovery Grant, and Weston Brain Institute awards to YIM, the *Brain Canada Foundation* and *Health Canada* support to the McConnell Brain Imaging Center at the Montreal Neurological Institute, and the *European Union's Horizon 2020 Framework Programme for Research and Innovation* under the Specific Grant Agreements 785907 (Human Brain Project SGA2) and 945539 (Human Brain Project SGA3) awarded to NPG and KZ. Multi-modal imaging and clinical data collection and sharing for this project was funded by PPMI (www.ppmi-info.org/data). A public-private partnership, PPMI is funded by the Michael J. Fox Foundation for Parkinson's Research and funding partners, including AbbVie, Allergan, Amathus Therapeutics, Avid Radiopharmaceuticals, Biogen, BioLegend, Bristol Myers Squibb, Celgene, Denali Therapeutics, GE Healthcare, Genentech, GlaxoSmithKline plc., Golub Capital, Handl Therapeutics, Insitro, Janssen Neuroscience, Eli Lilly and Company, Lundbeck, Merck Sharp & Dohme Corp., Meso Scale Discovery, Neurocrine Biosciences, Pfizer Inc., Piramal Group, Prevail Therapeutics, Roche, Sanofi Genzyme, Servier Laboratories, Takeda Pharmaceutical Company Limited, Teva Pharmaceutical Industries Ltd., UCB, Verily Life Sciences, and Voyager Therapeutics Inc. The full list of funding partners is available at www.ppmi-info.org/about-ppmi/who-we-are/study-sponsors.

## Author contributions

A.F.K. helped conceptualize the project, preprocessed the data, implemented the model, conducted the analysis, and wrote the paper. Q.A. and S.-J.L. helped preprocess the data. T.R.B. and F.C. contributed statistical methods. Y.Z. helped preprocess the data. N.P.-G. collected the receptor data. Y.I.-M. conceptualized and supervised the project. All authors helped interpret the results and revise the paper.

## Competing interests

The authors declare no competing interests.
