## [Peer Review file · Nature Communications]

Reviewers' comments:

Reviewer #1 (Remarks to the Author):

This paper combines data on receptor density from postmortem autoradiography (15 different receptors) with longitudinal neuroimaging data in PD with the aim to (a) identify patient-specific alterations, (b) perform causal modeling to specify potential mechanistic changes across patients, and (c) map individual brain regions in which specific receptor types influence PD-related neurodegeneration. The structure of the current study is similar to that of a previous paper in AD (Brain 2022): the receptor maps are from the same four brains as in the earlier study, with the exception that the number of cytoarchitecturally defined cortical areas was increased from 44 to 57.

Here, the authors analyzed the longitudinal PPMI database with scans from 71 PD patients who underwent structural MRI, diffusion tensor MRI (DTI), and/or SPECT DAT binding studies at three or more time points. Spatial distribution maps were constructed using regional density templates.

For 15 receptors (from the glutamatergic, GABAergic, cholinergic, adrenergic, serotonergic, and dopaminergic families), anatomical connectivity was estimated from the high-resolution Human Connectome Project template. Receptor-informed generative computational models were used to evaluate longitudinal changes in six biological factors (gray matter density (GM), fractional amplitude of low frequency fluctuations (fALFF), dopamine transporter SPECT, fractional anisotropy, mean diffusivity, and dendrite density) for each participant. The authors found that the receptor maps significantly improved the fitting of multifactorial neurodegeneration data over time in PD patients. They also show that motor/psychomotor symptoms and visuospatial dysfunction in PD are associated respectively with distinct receptor patterns (principal components), and that PD neurodegeneration is associated with model-derived changes in receptor influence involving different brain receptors. The results are interesting in that the interaction between the imaging and the receptor maps explains more of the variance in individual patient time courses than the imaging alone.

General comments:

1. While interesting, the study relies on using normal brains to evaluate receptor-mediated interactions with pathology, and the imaging measures (e.g., no mm loss, connectivity changes) are themselves indirect descriptors of the underlying disease process. Although a similar approach has been used to explain pathological changes in other diseases, the causal basis of the observations needs to be better explained.
2. The authors do state that the D1 receptor map is not a stable predictor of the DAT alterations. This is perhaps not surprising given that D2 receptor maps are (likely important) are not included in the model. While the approach is elegant, it is not clear what specifically was learned about PD. That many transmitter systems are affected in different sets of brain regions is known through earlier imaging studies.

Specific Comments:

1. Data from healthy subjects were not included in the study. How did the authors account for potential confounding effects of healthy aging in the assessment of within-and between-subject differences over time?
2. Why was it that including the receptor map did not improve the model of undirected microstructural damage (MD)?
3. Figures 5A and B: the x-axis is labelled as "Latent projection of clinical scores." This needs to be better explained. In Methods, the authors state: "We calculated cognitive decline as the linear best fit rate of change of each cognitive score with respect to examination date. Thus, for each patient, symptomatic decline was represented by a set of 11 rates of change." How were the 11 rates of change used in the correlation analyses with biological mechanisms, represented as the "Latent projection of clinical scores" shown in these figures? (2) Why not use the actual rate of changes of the clinical scores, instead of the fitted rate of change (which assumes that the changes are linear over time in every subject)?
4. How is the magnitude of receptor influence on each imaging modality reflected in the data shown in Figure 7B?
5. How do the authors plan to validate the current finding?

Reviewer #2 (Remarks to the Author):

Remarks to authors

The study presented has several interesting results. The authors make a compelling case for the need to use data from different modalities by showing how receptor enriched analysis increases explained variance. It is indeed crucial to examine different morphometrics that represent different components of ageing and disease, and different imaging modalities, in order to understand neurodegeneration and its progression. Now there is a wealth of atlases and public datasets available and it is crucial to make the most of these rich datasets. This is particularly important for those disease such as Parkinson's, where there is the need to take into account not only the different types of underlying protein pathology but also their possible different interactions. The results of the study are consistent with previous literature showing that different receptors are involved in different domains/features of Parkinson's disease and expand previous findings: in particular, the use of different biological factors the authors use to examine explicitly different mechanisms. In fact, using patient-specific generative brain modelling the authors identify receptors involved in different mechanisms of PD progression – the authors do not use true personalised longitudinal receptor data, but by using model weights of the specific mechanisms that are mediated by such receptors, they can model individualised trajectories. Of particular interest are the results showing the two components of PD (PC1 being more pertaining to the motor domain, PC2 to the cognitive-psychiatric-mood domain), which is consistent with the literature, and in particular how different mechanisms related to these two different clusters, highlighting the complexity associated to Parkinson's disease. Only by understanding the extent of this complexity it will be possible to tackle the broad range of symptoms associated to the disease and to monitor and address its heterogenous progression.

The receptors density extraction methods are sound and consistent with previous work using the same donors (e.g. Zachlod, D., Bludau, S., Cichon, S., Palomero-Gallagher, N., & Amunts, K. (2022). Combined analysis of cytoarchitectonic, molecular and transcriptomic patterns reveal differences in brain organization across human functional brain systems. *Neuroimage*, 257, 119286 and Khan, Ahmed Faraz, et al. "Personalized brain models identify neurotransmitter receptor changes in Alzheimer's disease." *Brain* 145.5 (2022): 1785-1804).

The use of quantitative in vitro receptor autoradiography makes the spatial resolution excellent. It is remarkable the use of age congruent donors, of which there is increasing need when studying relationship between brain structural features and receptor density in elderly adults and in particular patients with neurodegenerative disorders. This allows to have a better representation of non morbid receptor density at the age range consistent with that of patients.

The mention that future work will extend the presented results with voxel-scale whole brain receptor maps is exciting.

Questions for the authors:

- Data harmonisation is carried out using an empirical Bayes method, ComBat, that has been validated in several publications, is the harmonisation method of choice of the ENIGMA group and can take biological variance into account and correcting only for scanner/site variance. Could the authors please specify which biological covariates (those whereby variance was retained by the model) have been specified in the model - it was age and gender or were other clinical variables included (e.g. disease duration at the time of the scan/medication/.. etc)? The authors write "to correct for site and scanner effects". As the scanner/site-specific scaling factor is usually one, it would be perhaps clearer if the authors clarified that they used scanner (or site, as preferred) for this purpose and age, gender etc as biological covariates.

- The authors consider several measures of PD progression making the results very interesting and covering different relevant microstructural integrity measures. Being this a longitudinal analysis, was there a specific choice behind not using local gyrification index - literature has shown that it proves particularly useful in revealing different features in PD staging and in particular giving an index of Lewy body pathology, as for instance described in:

Sterling, N. W., Wang, M., Zhang, L., Lee, E. Y., Du, G., Lewis, M. M., ... & Huang, X. (2016). Stage-dependent loss of cortical gyrification as Parkinson's disease "unfolds". *Neurology*, 86(12), 1143-1151 where the authors show that gyrification is stage-dependent and associated with PD progression, continuing to decline in data after 5 years. This has implications as LG may possibly not reflect Lewy pathology directly but other processes, or that Lewy pathology might be correlated with cell death but not equally across cortical regions/or associated to a layer-specific pattern of pathology/changes in underlying white matter.

- The model seems to take into account spatial autocorrelation by computing a null distribution using randomly permuted receptor maps (for a comprehensive discussion see Markello, R. D., & Misic, B. (2021). Comparing spatial null models for brain maps. *NeuroImage*, 236, 118052). The authors may wish to add a note about this to make this clearer. As the use of multiple brain maps entails spatial autocorrelation and this is something that has been start being discussed in greater detail in the past couple of years, it would add to the methods to mention that this was taken care of.

- Having never used data from Düsseldorf donor program, this question might be less relevant. However, as the Allen Brain Atlas has some idiosyncrasies that increase the variability of the data, for example only two of the individuals have bi-hemispheric samples, or the samples of brain areas were obtained with different stereotactic coordinates; are the donors from which the receptor density information is extracted characterised by high inter-individual variability and if so did the authors take measures to keep this to a minimum? Did all the participants in this study have data from both hemispheres? If not, could the authors please elaborate on this and whether any measures have been taken in order to account for that? (for instance a subject-specific parcellation for each donor? as done for example in [doi:10.1093/brain/awy252](https://doi.org/10.1093/brain/awy252), where the Allen Brain Atlas was used).

Reviewers' comments:

Reviewer #1 (Remarks to the Author):

This paper combines data on receptor density from postmortem autoradiography (15 different receptors) with longitudinal neuroimaging data in PD with the aim to (a) identify patient-specific alterations, (b) perform causal modeling to specify potential mechanistic changes across patients, and (c) map individual brain regions in which specific receptor types influence PD-related neurodegeneration. The structure of the current study is similar to that of a previous paper in AD (Brain 2022): the receptor maps are from the same four brains as in the earlier study, with the exception that the number of cytoarchitectonically defined cortical areas was increased from 44 to 57.

Here, the authors analyzed the longitudinal PPMI database with scans from 71 PD patients who underwent structural MRI, diffusion tensor MRI (DTI), and/or SPECT DAT binding studies at three or more time points. Spatial distribution maps were constructed using regional density templates.

For 15 receptors (from the glutamatergic, GABAergic, cholinergic, adrenergic, serotonergic, and dopaminergic families), anatomical connectivity was estimated from the high-resolution Human Connectome Project template. Receptor-informed generative computational models were used to evaluate longitudinal changes in six biological factors (gray matter density (GM), fractional amplitude of low frequency fluctuations (fALFF), dopamine transporter SPECT, fractional anisotropy, mean diffusivity, and dendrite density) for each participant. The authors found that the receptor maps significantly improved the fitting of multifactorial neurodegeneration data over time in PD patients. They also show that motor/psychomotor symptoms and visuospatial dysfunction in PD are associated respectively with distinct receptor patterns (principal components), and that PD neurodegeneration is associated with model-derived changes in receptor influence involving different brain receptors. The results are interesting in that the interaction between the imaging and the receptor maps explains more of the variance in individual patient time courses than the imaging alone.

General comments:

1. While interesting, the study relies on using normal brains to evaluate receptor-mediated interactions with pathology, and the imaging measures (e.g., no mm loss, connectivity changes) are themselves indirect descriptors of the underlying disease process. Although a similar approach has been used to explain pathological changes in other diseases, the causal basis of the observations needs to be better explained.
2. The authors do state that the D1 receptor map is not a stable predictor of the DAT alterations. This is perhaps not surprising given that D2 receptor maps are (likely important) are not included in the model. While the approach is elegant, it is not clear what specifically was learned about PD. That many transmitter systems are affected in different sets of brain regions is known through earlier imaging studies.

Specific Comments:

1. Data from healthy subjects were not included in the study. How did the authors account for potential confounding effects of healthy aging in the assessment of within-and between-subject differences over time?
2. Why was it that including the receptor map did not improve the model of undirected microstructural damage (MD)?
3. Figures 5A and B: the x-axis is labelled as “Latent projection of clinical scores.” This needs to be better explained. In Methods, the authors state: “We calculated cognitive decline as the linear best fit rate of change of each cognitive score with respect to examination date. Thus, for each patient, symptomatic decline was represented by a set of 11 rates of change.” How were the 11 rates of change used in the correlation analyses with biological mechanisms, represented as the “Latent projection of clinical scores” shown in these figures? (2) Why not use the actual rate of changes of the clinical scores, instead of the fitted rate of change (which assumes that the changes are linear over time in every subject)?
4. How is the magnitude of receptor influence on each imaging modality reflected in the data shown in Figure 7B?
5. How do the authors plan to validate the current finding?

Reviewer #2 (Remarks to the Author):

Remarks to authors

The study presented has several interesting results. The authors make a compelling case for the need to use data from different modalities by showing how receptor enriched analysis increases explained variance. It is indeed crucial to examine different morphometrics that represent different components of ageing and disease, and different imaging modalities, in order to understand neurodegeneration and its progression. Now there is a wealth of atlases and public datasets available and it is crucial to make the most of these rich datasets. This is particularly important for those disease such as Parkinson's, where there is the need to take into account not only the different types of underlying protein pathology but also their possible different interactions. The results of the study are consistent with previous literature showing that different receptors are involved in different domains/features of Parkinson's disease and expand previous findings: in particular, the use of different biological factors the authors use to examine explicitly different mechanisms. In fact, using patient-specific generative brain modelling the authors identify receptors involved in different mechanisms of PD progression – the authors do not use true personalised longitudinal receptor data, but by using model weights of the specific mechanisms that are mediated by such receptors, they can model individualised trajectories. Of particular interest are the results showing the two components of PD (PC1 being more pertaining to the motor domain, PC2 to the cognitive-psychiatric-mood domain), which is consistent with the literature, and in particular how different mechanisms related to these two different clusters, highlighting the complexity associated to Parkinson's disease. Only by understanding the extent of this complexity it will be possible to tackle the broad range of symptoms associated to the disease and to monitor and address its heterogenous progression.

The receptors density extraction methods are sound and consistent with previous work using the same donors (e.g. Zachlod, D., Bludau, S., Cichon, S., Palomero-Gallagher, N., & Amunts, K. (2022). Combined analysis of cytoarchitectonic, molecular and transcriptomic patterns reveal differences in brain organization across human functional brain systems. *Neuroimage*, 257, 119286 and Khan, Ahmed Faraz, et al. "Personalized brain models identify neurotransmitter receptor changes in Alzheimer's disease." *Brain* 145.5 (2022): 1785-1804).

The use of quantitative in vitro receptor autoradiography makes the spatial resolution excellent. It is remarkable the use of age congruent donors, of which there is increasing need when studying relationship between brain structural features and receptor density in elderly adults and in particular patients with neurodegenerative disorders. This allows to have a better representation of non morbid receptor density at the age range consistent with that of patients. The mention that future work will extend the presented results with voxel-scale whole brain

receptor maps is exciting.

Questions for the authors:

- Data harmonisation is carried out using an empirical Bayes method, ComBat, that has been validated in several publications, is the harmonisation method of choice of the ENIGMA group and can take biological variance into account and correcting only for scanner/site variance. Could the authors please specify which biological covariates (those whereby variance was retained by the model) have been specified in the model - it was age and gender or were other clinical variables included (e.g. disease duration at the time of the scan/medication/.. etc)? The authors write “to correct for site and scanner effects”. As the scanner/site-specific scaling factor is usually one, it would be perhaps clearer if the authors clarified that they used scanner (or site, as preferred) for this purpose and age, gender etc as biological covariates.

- The authors consider several measures of PD progression making the results very interesting and covering different relevant microstructural integrity measures. Being this a longitudinal analysis, was there a specific choice behind not using local gyrification index - literature has shown that it proves particularly useful in revealing different features in PD staging and in particular giving an index of Lewy body pathology, as for instance described in:

Sterling, N. W., Wang, M., Zhang, L., Lee, E. Y., Du, G., Lewis, M. M., ... & Huang, X. (2016). Stage-dependent loss of cortical gyrification as Parkinson’s disease “unfolds”. *Neurology*, 86(12), 1143-1151 where the authors show that gyrification is stage-dependent and associated with PD progression, continuing to decline in data after 5 years. This has implications as LG may possibly not reflect Lewy pathology directly but other processes, or that Lewy pathology might be correlated with cell death but not equally across cortical regions/or associated to a layer-specific pattern of pathology/changes in underlying white matter.

- The model seems to take into account spatial autocorrelation by computing a null distribution using randomly permuted receptor maps (for a comprehensive discussion see Markello, R. D., & Misić, B. (2021). Comparing spatial null models for brain maps. *NeuroImage*, 236, 118052). The authors may wish to add a note about this to make this clearer. As the use of multiple brain maps entails spatial autocorrelation and this is something that has been start being discussed in greater detail in the past couple of years, it would add to the methods to mention that this was taken care of.

- Having never used data from Düsseldorf donor program, this question might be less relevant. However, as the Allen Brain Atlas has some idiosyncrasies that increase the variability of the data, for example only two of the individuals have bi-hemispheric samples, or the samples of brain areas were obtained with different stereotactic coordinates; are the donors from which the

receptor density information is extracted characterised by high inter-individual variability and if so did the authors take measures to keep this to a minimum? Did all the participants in this study have data from both hemispheres? If not, could the authors please elaborate on this and whether any measures have been taken in order to account for that? (for instance a subject-specific parcellation for each donor? as done for example in doi:10.1093/brain/awy252, where the Allen Brain Atlas was used).

Responses to reviewers' comments:

Reviewer #1 (Remarks to the Author):

This paper combines data on receptor density from postmortem autoradiography (15 different receptors) with longitudinal neuroimaging data in PD with the aim to (a) identify patient-specific alterations, (b) perform causal modeling to specify potential mechanistic changes across patients, and (c) map individual brain regions in which specific receptor types influence PD-related neurodegeneration. The structure of the current study is similar to that of a previous paper in AD (Brain 2022): the receptor maps are from the same four brains as in the earlier study, with the exception that the number of cytoarchitecturally defined cortical areas was increased from 44 to 57.

Here, the authors analyzed the longitudinal PPMI database with scans from 71 PD patients who underwent structural MRI, diffusion tensor MRI (DTI), and/or SPECT DAT binding studies at three or more time points. Spatial distribution maps were constructed using regional density templates.

For 15 receptors (from the glutamatergic, GABAergic, cholinergic, adrenergic, serotonergic, and dopaminergic families), anatomical connectivity was estimated from the high-resolution Human Connectome Project template. Receptor-informed generative computational models were used to evaluate longitudinal changes in six biological factors (gray matter density (GM), fractional amplitude of low frequency fluctuations (fALFF), dopamine transporter SPECT, fractional anisotropy, mean diffusivity, and dendrite density) for each participant. The authors found that the receptor maps significantly improved the fitting of multifactorial neurodegeneration data over time in PD patients. They also show that motor/psychomotor symptoms and visuospatial dysfunction in PD are associated respectively with distinct receptor patterns (principal components), and that PD neurodegeneration is associated with model-derived changes in receptor influence involving different brain receptors. The results are interesting in that the interaction between the imaging and the receptor maps explains more of the variance in individual patient time courses than the imaging alone.

Response: We would like to thank the reviewer for taking the time to assess our manuscript. The comments have helped us clarify several points in the manuscript, which we have also described in the specific responses below. We have also changed the title of the manuscript to “*Individual differences in neurotransmitter receptor mechanisms underlie co-occurring motor and visuospatial disease axes in Parkinson’s disease*”, to highlight the main result showing multi-receptor association with distinct symptomatic profiles.

General comments:

Comment 1. *While interesting, the study relies on using normal brains to evaluate receptor-mediated interactions with pathology, and the imaging measures (e.g., no mm loss, connectivity changes) are themselves indirect descriptors of the underlying disease process. Although a similar approach has been used to explain pathological changes in other diseases, the causal basis of the observations needs to be better explained.*

Response: We appreciate the opportunity to clarify our approach. The usage of healthy aged receptor templates in combination with neuroimaging measures from PD patients is a key point in our work. Firstly, in vivo, whole brain, high spatial resolution, multi-receptor data is simply not available for PD patients. We note that the different receptor maps are not very correlated with each other (Supplementary Fig. S1-S6) and the “multi-receptor fingerprint” of each (cyto-architectonically defined) brain region is distinct (Zilles & Palomero-Gallagher, 2017, <https://doi.org/10.3389/fnana.2017.00078>). This suggests that receptor distributions differentiate regions and layers within the brain, potentially reflecting regional susceptibility to different forms of neurodegeneration (e.g. excitotoxicity, vascular alterations, localized cell death and atrophy, etc.). Furthermore, these autoradiography-derived receptor maps from healthy aged subjects seem to have physiological relevance to a broad population, for example correlating well with drug-induced changes to cerebral blood flow in healthy young adults based on specific drug-receptor affinity (Dukart et al., 2018, <https://doi.org/10.1038/s41598-018-22444-0>). Healthy neurotransmitter receptor and transporter templates from PET and SPECT imaging also spatially correlate with PD rsfMRI alterations (Dukart et al., 2020, <https://doi.org/10.1002/hbm.25244>), brain regions with dyskinesia- and parkinsonism-associated atrophy in schizophrenia patients (Sakrieda et al., 2022,

<https://doi.org/10.1093/braincomms/fcac190>), gray matter volume changes in symptomatic and prodromal genetic subtypes of FTD (Premi et al., 2023, <https://doi.org/10.1016/j.nbd.2023.106068>), and functional alterations in behavioral variant FTD (Hahn et al., 2023 - preprint, <https://doi.org/10.1101/2022.08.30.22278624>). Furthermore, in vitro multi-receptor autoradiography of the caudate nucleus and midcingulate area 24 of progressive supranuclear palsy (PSP) patients showed differentiation of autopsied patients from age-matched controls, as well as diverging alterations in clinical subgroups of PSP (Chiu et al., 2017, <https://doi.org/10.1186/s13195-017-0259-5>). In this related movement disorder, notable, previously unknown receptor alterations (to kainate and adenosine type 1 receptors) were discovered, supporting the case for more thorough receptor mapping studies in neurodegenerative disorders. Complementing such analyses, in our work, we attempted to use individualized, in silico modeling to quantify the roles of receptors in mediating interactions between imaging-measured processes, regional susceptibility to neurodegeneration, and inter-individual symptomatic variability.

Although neuroimaging markers are indirect measures of underlying disease processes, the modalities considered in this work can be sensitive to disease progression particularly in the early stages (e.g. Mitchell et al., 2021, <https://doi.org/10.1001/jamaneurol.2021.1312>). Furthermore, while neuroimaging alone can uncover certain structural and functional aspects of disease progression via longitudinal analysis, the latent molecular basis cannot be inferred without mechanistic modeling. Using individualized model parameters to infer molecular mechanisms neurodegenerative is a nascent and promising approach (e.g. Stefanovski et al., 2019, <https://doi.org/10.3389/fncom.2019.00054>). Furthermore, other studies have also found improved correspondence between personalized model-derived parameters and clinical phenotype in Alzheimer's disease (Zimmermann et al., 2018, <https://doi.org/10.1016/j.nicl.2018.04.017>).

We used the 15 receptor distribution template maps analogously to a prior distribution in our model. Each patient's neuroimaging models are fit independently, and the optimized model weights reflect inter-patient differences in the role of a given receptor for a particular neuroimaging modality. The causal modeling approach allows us to disentangle direct effects of

neuroimaging changes from receptor-mediated interactions influencing the progression of pathological structural, functional, and dopaminergic neuroimaging alterations. For example, the gray matter atrophy model of patient A may show a stronger NMDA contribution (i.e. large model weights of NMDA terms) compared to patient B, suggesting that this glutamatergic receptor-driven susceptibility to atrophy may be at play in patient A but not patient B. Consequently, statistical stability tests, as well as robust and significant multivariate correlation with symptoms allows us to infer which mechanisms are associated with disease progression. We have expanded on the description of our methods in the second and third discussion paragraphs.

Comment 2. *The authors do state that the D1 receptor map is not a stable predictor of the DAT alterations. This is perhaps not surprising given that D2 receptor maps are (likely important) are not included in the model. While the approach is elegant, it is not clear what specifically was learned about PD. That many transmitter systems are affected in different sets of brain regions is known through earlier imaging studies.*

Response: We appreciate the reviewer giving us the opportunity to clarify our main results. By linking spatial distribution templates of 15 important neurotransmitter receptors from neurologically healthy aged subjects with multi-modal, longitudinal neuroimaging (including structural, functional and diffusion MRI, and dopaminergic SPECT), we have developed the first computational model inferring specific pathways of neurotransmitter-mediated neurodegeneration in PD. At a time when the field is increasingly acknowledging the multi-system nature of PD, we show that there are two co-occurring “disease axes” with distinct symptom profiles and underlying mechanistic basis in multiple receptor-mediated interactions in a cohort of 71 PD patients. For example, we demonstrate that cholinergic receptor influence on neuroimaging changes appears to be the main driving factor for visuospatial and cognitive symptoms.

Our manuscript addresses critical, unanswered questions in PD literature: i) we uncover a regional susceptibility to neurodegeneration based on receptor expression (Fig. 3 & 4), ii) we quantify the receptor basis of interactions between pathological processes (e.g. glutamatergic microstructural damage, cholinergic and GABAergic interaction with dopaminergic transporter

loss; Fig. 3 & 4), iii) we demonstrate that inter-individual variability in these interactions is strongly related to clinical heterogeneity via 2 co-occurring (motor & non-motor) disease axes (Fig. 5 & 6), and iv) also for the first time, we determine specific regional influence maps for each receptor and pathological process. By providing specific pathways of interaction via our model (e.g. cholinergic effects on t1/t2 ratio changes correlating to the secondary clinical component), we lay the modeling foundations for future work in designing personalized therapeutic interventions to target specific molecular pathways in PD.

Reviewer 1's specific comments:

Comment 3. *Data from healthy subjects were not included in the study. How did the authors account for potential confounding effects of healthy aging in the assessment of within-and between-subject differences over time?*

Response: We thank the reviewer for raising this extremely pertinent point. As an inclusion criterion in our work, all subjects were required to have at least 3 imaging and clinical visits to facilitate personalized longitudinal modeling. This criterion disqualified all healthy controls, who generally had less imaging data in the PPMI dataset. However, PD is primarily defined and diagnosed by clinical symptoms. It is safe to assume that healthy control subjects would not display rapidly deteriorating motor, cognitive, or psychiatric symptoms that are associated with PD. As such, we considered our model-derived mechanisms to be associated with PD rather than healthy aging when their correlation with multiple PD-related clinical assessments was statistically stable and significant. This allows us to account for neuroimaging alterations that may be associated with “healthy aging” (i.e. non-symptomatic), both within and between subjects. The multivariate correlation method is described methodologically in the subsection “Statistical analysis: Covariance of biological mechanisms with clinical symptoms”, and the results are presented in the subsection “Two axes of receptor-pathology alterations underlie clinical symptoms in PD”. We have clarified this use of the clinical data to disentangle disease-related mechanisms in a new paragraph (paragraph 15) in the discussion section.

Comment 4. *Why was it that including the receptor map did not improve the model of undirected microstructural damage (MD)?*

Response: We appreciate the reviewer pointing out the varying relevance of receptor maps to different neuroimaging-derived models. Our work aimed to ascertain the extent to which receptor expression can explain individualized neurodegenerative changes as measured by different neuroimaging modalities. We note that all imaging modalities are improved by the inclusion of receptor maps, explaining an average of 42.3% more variance (Fig. 2a,b). For most modalities, this improvement is significant when corrected for larger model size in 93%-100% of subjects (Fig. 2c) and based on a permutation analysis with null receptor distributions in 93%-99% of subjects (Fig. 2d). However, for mean diffusivity (MD), the improvement is significant in only 80% of subjects when corrected for model size, and over null receptor maps in approximately two-thirds of patients. As mentioned in Paragraph 4 of the discussion section, “while diffusion MRI can be sensitive to aspects of gray matter microstructure, it is less accurate than in white matter due to the heterogeneity of tissues and their (lack of) organization”. In our gray matter ROIs, defined by receptor- and cyto-architecture, partial volume effects may preferentially affect the MD models.

Comment 5. *Figures 5A and B: the x-axis is labelled as “Latent projection of clinical scores.” This needs to be better explained. In Methods, the authors state: “We calculated cognitive decline as the linear best fit rate of change of each cognitive score with respect to examination date. Thus, for each patient, symptomatic decline was represented by a set of 11 rates of change.” How were the 11 rates of change used in the correlation analyses with biological mechanisms, represented as the “Latent projection of clinical scores” shown in these figures? (2) Why not use the actual rate of changes of the clinical scores, instead of the fitted rate of change (which assumes that the changes are linear over time in every subject)?*

Response: We appreciate the opportunity to clarify the results presented in Figures 5a and 5b. For each subject, we have a set of model weights, representing the influence of different model-inferred biological mechanisms, and a set of clinical assessments at each visit. There are varying numbers of clinical visits at varying times for each subject, and to unify this data into a single variable while reducing noise, we calculate the linear rate of change of each clinical assessment over all visits. As described in the subsection “Statistical analysis: Covariance of biological mechanisms with clinical symptoms”, we then performed a multivariate partial least squares

(PLS) analysis, based on a singular value decomposition (SVD) of the cross-covariance between the subjects' model weights and their multivariate clinical assessments' rates of change. By doing this, we project both the multiple model weights and multiple clinical rates of decline into orthonormal latent bases. These projections are fit in a way that maximizes the correlation between the projected weights and projected rates of decline. As such, the x-axes of Figs. 5a and 5b show the projected clinical assessment rates of decline in the first 2 latent axes, while the y-axes show the corresponding projected model weights. We will include a clarification in the updated manuscript.

Comment 6. *How is the magnitude of receptor influence on each imaging modality reflected in the data shown in Figure 7B?*

Response: We thank the reviewer for pointing out the unclear presentation of Figure 7. This figure shows receptor influence maps, which quantify the informativeness of a particular receptor distribution in a given neuroimaging model. The color represents the effect size, which is the Wilcoxon statistic of the improvement in model residuals at each brain region due to the inclusion of the receptor map for this modality. Only significant regions are shown (based on a z-score compared to a null distribution of Wilcoxon statistics due to permuted receptor maps). The updated manuscript includes colormaps for the figure as well as an explicit reference to the colour representing the Wilcoxon statistic effect size in the figure legend.

Comment 7. *How do the authors plan to validate the current finding?*

Response: We appreciate the reviewer raising this question about future validation of the model. Using observational data, we plan to test the associations of model-derived receptor mechanisms with known drug intake data (e.g., the effects of cholinergic medication intake on model-derived cholinergic mechanisms). Additionally, transcriptomic and autoradiographic analysis of post-mortem tissues from brain banks are another avenue for validation. Furthermore, similar receptor expression data is available for animal models, where more invasive validation can be conducted. More extensive validation of the presented model is an important step in our future research efforts, but outside the scope of this manuscript.

Reviewer #2 (Remarks to the Author):

Remarks to authors

The study presented has several interesting results. The authors make a compelling case for the need to use data from different modalities by showing how receptor enriched analysis increases explained variance. It is indeed crucial to examine different morphometrics that represent different components of ageing and disease, and different imaging modalities, in order to understand neurodegeneration and its progression. Now there is a wealth of atlases and public datasets available and it is crucial to make the most of these rich datasets. This is particularly important for those diseases such as Parkinson's, where there is the need to take into account not only the different types of underlying protein pathology but also their possible different interactions. The results of the study are consistent with previous literature showing that different receptors are involved in different domains/features of Parkinson's disease and expand previous findings: in particular, the use of different biological factors the authors use to examine explicitly different mechanisms. In fact, using patient-specific generative brain modelling the authors identify receptors involved in different mechanisms of PD progression – the authors do not use true personalised longitudinal receptor data, but by using model weights of the specific mechanisms that are mediated by such receptors, they can model individualised trajectories. Of particular interest are the results showing the two components of PD (PC1 being more pertaining to the motor domain, PC2 to the cognitive-psychiatric-mood domain), which is consistent with the literature, and in particular how different mechanisms related to these two different clusters, highlighting the complexity associated to Parkinson's disease. Only by understanding the extent of this complexity it will be possible to tackle the broad range of symptoms associated to the disease and to monitor and address its heterogeneous progression.

*The receptors density extraction methods are sound and consistent with previous work using the same donors (e.g. Zachlod, D., Bludau, S., Cichon, S., Palomero-Gallagher, N., & Amunts, K. (2022). Combined analysis of cytoarchitectonic, molecular and transcriptomic patterns reveal differences in brain organization across human functional brain systems. *Neuroimage*, 257, 119286 and Khan, Ahmed Faraz, et al. "Personalized brain models identify neurotransmitter receptor changes in Alzheimer's disease." *Brain* 145.5 (2022): 1785-1804).*

The use of quantitative in vitro receptor autoradiography makes the spatial resolution excellent. It is remarkable the use of age congruent donors, of which there is increasing need when studying relationship between brain structural features and receptor density in elderly adults and in particular patients with neurodegenerative disorders. This allows to have a better representation of non morbid receptor density at the age range consistent with that of patients. The mention that future work will extend the presented results with voxel-scale whole brain receptor maps is exciting.

Response: We would like to thank the reviewer for their feedback on our manuscript. Particularly, we have clarified several methodological points in the manuscript, which are also addressed in the responses to the specific comments below.

Questions for the authors:

Comment 1. *Data harmonisation is carried out using an empirical Bayes method, ComBat, that has been validated in several publications, is the harmonisation method of choice of the ENIGMA group and can take biological variance into account and correcting only for scanner/site variance. Could the authors please specify which biological covariates (those whereby variance was retained by the model) have been specified in the model - it was age and gender or were other clinical variables included (e.g. disease duration at the time of the scan/medication/.. etc)? The authors write “to correct for site and scanner effects”. As the scanner/site-specific scaling factor is usually one, it would be perhaps clearer if the authors clarified that they used scanner (or site, as preferred) for this purpose and age, gender etc as biological covariates.*

Response: We appreciate the reviewer for giving us the opportunity to clarify the harmonization procedure. We harmonized the imaging data using ComBat with the site as a variable (each site used the same scanner for all our subjects), while using diagnosis, age, education level, sex, and (left or right) handedness as biological covariables to preserve. We have clarified these variables in the updated manuscript (*Methods: Data description and processing: multimodal neuroimaging data fusion*).

Comment 2. *The authors consider several measures of PD progression making the results very*

interesting and covering different relevant microstructural integrity measures. Being this a longitudinal analysis, was there a specific choice behind not using local gyrification index - literature has shown that it proves particularly useful in revealing different features in PD staging and in particular giving an index of Lewy body pathology, as for instance described in:

*Sterling, N. W., Wang, M., Zhang, L., Lee, E. Y., Du, G., Lewis, M. M., ... & Huang, X. (2016). Stage-dependent loss of cortical gyrification as Parkinson's disease "unfolds". *Neurology*, 86(12), 1143-1151 where the authors show that gyrification is stage-dependent and associated with PD progression, continuing to decline in data after 5 years. This has implications as LG may possibly not reflect Lewy pathology directly but other processes, or that Lewy pathology might be correlated with cell death but not equally across cortical regions/or associated to a layer-specific pattern of pathology/changes in underlying white matter.*

Response: We appreciate the reviewer suggesting cortical gyrification, a measure conveying structural information that is not captured by our existing structural modalities. Given their importance to PD, we included the basal ganglia and thalamus in our model, precluding any cortical surface-based measures, as our model required identical features across brain regions. However, we will consider the gyrification index in future cortical applications of our model.

Comment 3. *The model seems to take into account spatial autocorrelation by computing a null distribution using randomly permuted receptor maps (for a comprehensive discussion see Markello, R. D., & Misic, B. (2021). Comparing spatial null models for brain maps. *NeuroImage*, 236, 118052). The authors may wish to add a note about this to make this clearer. As the use of multiple brain maps entails spatial autocorrelation and this is something that has been start being discussed in greater detail in the past couple of years, it would add to the methods to mention that this was taken care of.*

Response: We appreciate the reviewer giving us the opportunity to clarify the null model generation method. We used a simple, non-parametric permutation to generate a null distribution of receptor maps. We currently lack voxel-scale receptor maps and aimed to include subcortical regions in our model. These factors precluded any non-parametric, cortical surface rotation-

based methods, as well as parametrized models based on the voxel-scale autocorrelation of true receptor maps. We have clarified these points in the updated manuscript (*Discussion*).

Comment 4. *Having never used data from Düsseldorf donor program, this question might be less relevant. However, as the Allen Brain Atlas has some idiosyncrasies that increase the variability of the data, for example only two of the individuals have bi-hemispheric samples, or the samples of brain areas were obtained with different stereotactic coordinates; are the donors from which the receptor density information is extracted characterised by high inter-individual variability and if so did the authors take measures to keep this to a minimum? Did all the participants in this study have data from both hemispheres? If not, could the authors please elaborate on this and whether any measures have been taken in order to account for that? (for instance a subject-specific parcellation for each donor? as done for example in [doi:10.1093/brain/awy252](https://doi.org/10.1093/brain/awy252), where the Allen Brain Atlas was used).*

Response: We thank the reviewer for raising these points about integrating histological data from multiple donors. Both hemispheres from all donors were used (<https://doi.org/10.3389/fnana.2017.00078>), and we have clarified this in the manuscript. We also note that inter-subject variability in regional receptor expression is low among the donor brains whose data was used, especially compared to inter-layer or inter-region variability. An upcoming work by other authors will present a 3D reconstruction of the receptor maps registered to a common structural template (<https://doi.org/10.1101/2022.11.18.517039>).

REVIEWER COMMENTS

Reviewer #1 (Remarks to the Author):

The authors have provided detailed responses to my questions, in particular regarding healthy control data and their model of undirected microstructural damage. They also satisfactorily addressed the issues relating to rates of change (Fig. 5) and receptor influence maps (Fig. 7). They also revised the manuscript and updated the figures, accordingly.

Reviewer #2 (Remarks to the Author):

The authors have responded to my comments in an overall satisfactory way.

The work is rigorous and the methodology is sound. In a field where there are still limitations in studying neurochemistry associated to neurodegenerative disease without using invasive techniques, the approach taken here has the merit of exploring a wide range of neurotransmitters with a rigorous methodology and to take individual differences into account. Heterogeneity is particularly difficult to address in neurodegenerative disease and in Parkinson's whereby the breadth of non-motor symptoms associated to it is very significant and is thought to depend on the different underlying pathology and protein-protein interactions. The individualised generative models to the longitudinal alterations is a remarkable approach in this sense as it caters to PD's heterogeneity. This method if further developed can be useful to address also other disease characterised by heterogeneity.

The study also adds to other previous research using receptor density maps and how including this type of data can enhance our understanding of brain organisation in healthy and diseased brain. Individualised receptor data is not only expensive but invasive, especially if the scope is to explore a number of neurotransmitter receptors, and especially in people with a neurodegenerative condition. Thus it is crucial that we continue to work on developing optimal solutions to understand the neurochemistry associated to brain networks affected by such diseases.

The re-MCM method being soon released for other researchers to use is also promising.

Reviewer #3 (Remarks to the Author):

The authors present a study into the relative regional influence of PD-related brain change measured by MRI and SPECT on motor and cognitive measures, using control-derived receptor maps to create statistical models that relate regional receptor densities derived from the donor control brains to

imaging indices of PD pathology derived from PPMI MRI and Ioflupane (DAT-SPECT) imaging. They investigate the regional PD pathology from the PPMI sample to the putative underlying receptor maps. Lastly, they decompose the data to derive axes of clinical and imaging measures along different clinical axes reflecting motor and cognitive scores.

Putative mechanisms behind motor and cognitive change in PD have been studied by various groups, and some of these studies have used in-vivo nuclear methods to determine the neuroreceptor pathology of Parkinson's with and without dementia. The influence of dopaminergic decline on motor symptoms is well known and a mainstay of therapy, it also contributes to cognitive frontal dysfunction, eg. reduced word fluency. Cholinergic dysfunction is associated with visual and visuospatial dysfunction, as well as memory loss seen in PD dementia (see the Bohnen papers and related literature). The decline in receptor density in PD is not necessarily predicted by their pre-morbid density. There are well-known anterior-posterior gradients in both cholinergic and dopaminergic progression of pathology in PD, and these are not reflected in the maps predicting neurodegeneration (cf. Fig 7 M2 and alpha4beta2, supp figure S3). Another issue with the dopamine methods is the lack of a cortical dopamine signal in the SPECT data. Ioflupane SPECT has too poor signal to noise to pick up the mesiofrontal cortical uptake seen with PET (and which is known to be affected by PD), which explains the lack of influence of D1 receptor density on the SPECT signal outside the basal ganglia (supp figure S6). Another issue inherent to the PPMI sample is that it represents a very mild, initial phenotype of Parkinson's, which is highly selective (close to diagnosis, remaining off medication) and not necessarily representative of the spectrum of PD. This means it may be more difficult to pick up an imaging signal outside of basal ganglia SPECT, compared to the atrophy seen in Alzheimer's and (often) in aMCI. The very low influence of the dopamine signal on the first, motor domain identified seems at odds with the biological basis of PD (cf. supp table S5), although this may be an artefact of the lack of a control group, and the specifics of the PPMI participants which presumably have lower variability of their motor scores than an unselected early PD sample.

While the authors are to be complimented on the work they have done modelling the influence of receptor expression on syndromatology, I am not convinced that the approach can inform us specifically about PD-related pathology which does not necessarily follow expression patterns of receptors. The exclusion of controls, and the distribution of the receptor influence maps make it difficult for me to understand if the correlations demonstrated are meaningful in terms of PD pathology, or simply correlative to the spatial distribution of pathology that may be driven by non-receptor related means, such as the propagation of alpha-synuclein pathology throughout the brain.

REVIEWER COMMENTS

Reviewer #1 (Remarks to the Author):

The authors have provided detailed responses to my questions, in particular regarding healthy control data and their model of undirected microstructural damage. They also satisfactorily addressed the issues relating to rates of change (Fig. 5) and receptor influence maps (Fig. 7). They also revised the manuscript and updated the figures, accordingly.

Reviewer #2 (Remarks to the Author):

The authors have responded to my comments in an overall satisfactory way.

The work is rigorous and the methodology is sound. In a field where there are still limitations in studying neurochemistry associated to neurodegenerative disease without using invasive techniques, the approach taken here has the merit of exploring a wide range of neurotransmitters with a rigorous methodology and to take individual differences into account. Heterogeneity is particularly difficult to address in neurodegenerative disease and in Parkinson's whereby the breadth of non-motor symptoms associated to it is very significant and is thought to depend on the different underlying pathology and protein-protein interactions. The individualised generative models to the longitudinal alterations is a remarkable approach in this sense as it caters to PD's heterogeneity. This method if further developed can be useful to address also other disease characterised by heterogeneity.

The study also adds to other previous research using receptor density maps and how including this type of data can enhance our understanding of brain organisation in healthy and diseased brain. Individualised receptor data is not only expensive but invasive, especially if the scope is to explore a number of neurotransmitter receptors, and especially in people with a neurodegenerative condition. Thus it is crucial that we continue to work on developing optimal solutions to understand the neurochemistry associated to brain networks affected by such diseases.

The re-MCM method being soon released for other researchers to use is also promising.

Reviewer #3 (Remarks to the Author):

The authors present a study into the relative regional influence of PD-related brain change measured by MRI and SPECT on motor and cognitive measures, using control-derived receptor maps to create statistical models that relate regional receptor densities derived from the donor control brains to imaging indices of PD pathology derived from PPMI MRI and Ioflupane (DAT-SPECT) imaging. They investigate the regional PD pathology from the PPMI sample to the

putative underlying receptor maps. Lastly, they decompose the data to derive axes of clinical and imaging measures along different clinical axes reflecting motor and cognitive scores.

Putative mechanisms behind motor and cognitive change in PD have been studied by various groups, and some of these studies have used in-vivo nuclear methods to determine the neuroreceptor pathology of Parkinson's with and without dementia. The influence of dopaminergic decline on motor symptoms is well known and a mainstay of therapy, it also contributes to cognitive frontal dysfunction, eg. reduced word fluency. Cholinergic dysfunction is associated with visual and visuospatial dysfunction, as well as memory loss seen in PD dementia (see the Bohnen papers and related literature). The decline in receptor density in PD is not necessarily predicted by their pre-morbid density. There are well-known anterior-posterior gradients in both cholinergic and dopaminergic progression of pathology in PD, and these are not reflected in the maps predicting neurodegeneration (cf. Fig 7 M2 and alpha4beta2, supp figure S3). Another issue with the dopamine methods is the lack of a cortical dopamine signal in the SPECT data. Ioflupane SPECT has too poor signal to noise to pick up the mesiofrontal cortical uptake seen with PET (and which is known to be affected by PD), which explains the lack of influence of D1 receptor density on the SPECT signal outside the basal ganglia (supp figure S6). Another issue inherent to the PPMI sample is that it represents a very mild, initial phenotype of Parkinson's, which is highly selective (close to diagnosis, remaining off medication) and not necessarily representative of the spectrum of PD. This means it may be more difficult to pick up an imaging signal outside of basal ganglia SPECT, compared to the atrophy seen in Alzheimer's and (often) in aMCI. The very low influence of the dopamine signal on the first, motor domain identified seems at odds with the biological basis of PD (cf. supp table S5), although this may be an artefact of the lack of a control group, and the specifics of the PPMI participants which presumably have lower variability of their motor scores than an unselected early PD sample.

While the authors are to be complimented on the work they have done modelling the influence of receptor expression on syndromatology, I am not convinced that the approach can inform us specifically about PD-related pathology which does not necessarily follow expression patterns of receptors. The exclusion of controls, and the distribution of the receptor influence maps make it difficult for me to understand if the correlations demonstrated are meaningful in terms of PD pathology, or simply correlative to the spatial distribution of pathology that may be driven by non-receptor related means, such as the propagation of alpha-synuclein pathology throughout the brain.

RESPONSES TO REVIEWER COMMENTS

Reviewer #1 (Remarks to the Author):

The authors have provided detailed responses to my questions, in particular regarding healthy control data and their model of undirected microstructural damage. They also satisfactorily addressed the issues relating to rates of change (Fig. 5) and receptor influence maps (Fig. 7). They also revised the manuscript and updated the figures, accordingly.

Response: We appreciate the reviewer's constructive suggestions and the opportunity to clarify these aspects in the manuscript.

Reviewer #2 (Remarks to the Author):

The authors have responded to my comments in an overall satisfactory way.

The work is rigorous and the methodology is sound. In a field where there are still limitations in studying neurochemistry associated to neurodegenerative disease without using invasive techniques, the approach taken here has the merit of exploring a wide range of neurotransmitters with a rigorous methodology and to take individual differences into account. Heterogeneity is particularly difficult to address in neurodegenerative disease and in Parkinson's whereby the breadth of non-motor symptoms associated to it is very significant and is thought to depend on the different underlying pathology and protein-protein interactions. The individualised generative models to the longitudinal alterations is a remarkable approach in this sense as it caters to PD's heterogeneity. This method if further developed can be useful to address also other disease characterised by heterogeneity.

The study also adds to other previous research using receptor density maps and how including this type of data can enhance our understanding of brain organisation in healthy and diseased brain. Individualised receptor data is not only expensive but invasive, especially if the scope is to explore a number of neurotransmitter receptors, and especially in people with a neurodegenerative condition. Thus it is crucial that we continue to work on developing optimal solutions to understand the neurochemistry associated to brain networks affected by such diseases.

The re-MCM method being soon released for other researchers to use is also promising.

Response: Thank you. We really appreciate the reviewer's positive comments and feedback.

Reviewer #3 (Remarks to the Author):

The authors present a study into the relative regional influence of PD-related brain change measured by MRI and SPECT on motor and cognitive measures, using control-derived receptor maps to create statistical models that relate regional receptor densities derived from the donor control brains to imaging indices of PD pathology derived from PPMI MRI and Ioflupane (DAT-SPECT) imaging. They investigate the regional PD pathology from the PPMI sample to the putative underlying receptor maps. Lastly, they decompose the data to derive axes of clinical and imaging measures along different clinical axes reflecting motor and cognitive scores.

Putative mechanisms behind motor and cognitive change in PD have been studied by various groups, and some of these studies have used in-vivo nuclear methods to determine the neuroreceptor pathology of Parkinson's with and without dementia. The influence of dopaminergic decline on motor symptoms is well known and a mainstay of therapy, it also contributes to cognitive frontal dysfunction, eg. reduced word fluency. Cholinergic dysfunction is associated with visual and visuospatial dysfunction, as well as memory loss seen in PD dementia (see the Bohnen papers and related literature). The decline in receptor density in PD is not necessarily predicted by their pre-morbid density. There are well-known anterior-posterior gradients in both cholinergic and dopaminergic progression of pathology in PD, and these are not reflected in the maps predicting neurodegeneration (cf. Fig 7 M2 and alpha4beta2, supp figure S3). Another issue with the dopamine methods is the lack of a cortical dopamine signal in the SPECT data. Ioflupane SPECT has too poor signal to noise to pick up the mesiofrontal cortical uptake seen with PET (and which is known to be affected by PD), which explains the lack of influence of D1 receptor density on the SPECT signal outside the basal ganglia (supp figure S6).

Another issue inherent to the PPMI sample is that it represents a very mild, initial phenotype of Parkinson's, which is highly selective (close to diagnosis, remaining off medication) and not necessarily representative of the spectrum of PD. This means it may be more difficult to pick up an imaging signal outside of basal ganglia SPECT, compared to the atrophy seen in Alzheimer's and (often) in aMCI. The very low influence of the dopamine signal on the first, motor domain identified seems at odds with the biological basis of PD (cf. supp table S5), although this may be an artefact of the lack of a control group, and the specifics of the PPMI participants which presumably have lower variability of their motor scores than an unselected early PD sample.

While the authors are to be complimented on the work they have done modelling the influence of receptor expression on syndromatology, I am not convinced that the approach can inform us specifically about PD-related pathology which does not necessarily follow expression patterns of receptors. The exclusion of controls, and the distribution of the receptor influence maps make it difficult for me to understand if the correlations demonstrated are meaningful in terms of PD pathology, or simply correlative to the spatial distribution of pathology that may be driven by

non-receptor related means, such as the propagation of alpha-synuclein pathology throughout the brain.

Response: We appreciate the reviewer's comments, particularly regarding the limitations of the imaging data in this PD cohort. The low cortical signal in DAT-SPECT imaging, the early and mild motor phenotype of the PPMI PD patients and the lack of controls are indeed limitations of our otherwise notably large, longitudinal, and uniquely multi-modal PD dataset. Nevertheless, we appreciate the opportunity to clarify and improve certain critical points about our modeling approach, the interpretations of our results, and how we have attempted to address the mentioned limitations.

General aspects:

The complexity of PD is increasingly acknowledged in terms of mixed pathology (Chu et al., 2023, <https://doi.org/10.1016/B978-0-323-85538-9.00012-2>), multi-neurotransmitter dysfunction (Titova et al., 2017, <https://doi.org/10.1007/s00702-016-1667-6>) and non-motor symptoms (Pfeiffer, 2016, <https://doi.org/10.1016/j.parkreldis.2015.09.004>). There is a critical need to develop novel methods that describe these complex, multi-factorial biological alterations, their underlying molecular mechanisms, and how they are associated with the spectrum of symptoms in PD (Espay et al. 2017, <https://doi.org/10.1002/mds.26913>). Our work is focused on this problem, rather than a case-control comparison of imaging between diagnostic classes.

We start by attempting to infer underlying mechanistic alterations. Specifically, we focus on interactions between altered biological factors that are partially mediated by neurotransmitter receptors. Given their critical roles in communication between cells and pharmacological response, neurotransmitter receptors are likely pathways by which different forms of neuropathology influence each other (Xu et al., 2012, <http://dx.doi.org/10.1016/j.pneurobio.2012.02.002>). For most receptors and pathological factors, this has not been characterized well in PD, partially due to a lack of in vivo receptor mapping. Importantly, we also consider other non-receptor mechanisms by which neuropathology (including gray matter trophy, resting state activity dysfunction, microstructural damage, etc.) evolves over disease progression, such as direct interactions between pathological factors and their inter-region propagation mediated by the structural connectome.

To characterize these biological mechanisms and their inter-individual differences, we use a model-based approach that combines healthy spatial distributions of 15 receptor types (as a-priori information) with imaging-derived metrics of 6 pathological factors from PPMI. For each patient, we fit causal models explaining the longitudinal alterations to each neuropathological factor. These models are interpretable, with each model weight representing a specific biological mechanism that is a mixture of effects due to the distributions of receptors and of imaging-measured pathology. Furthermore, these model weights, which are a set of variables unique to

each subject, also capture the subject-specific importance of receptor mechanisms. Although lacking individualized receptor maps, the templates provide a priori information about receptor distributions in healthy aged brains, and the model fitting procedure introduces inter-subject variability into the model weights of receptor terms representing their effects on imaging-measured pathology. Biologically, this may represent changes to the density of the receptor, or functional alterations involving its interactions.

We then robustly associate inter-individual variability in model weights with multi-domain clinical symptoms to find distinct axes of mechanistic-clinical co-variability. Since PD is primarily defined clinically, the prototypical healthy subject would be free from the range of (motor and non-motor) symptoms associated with PD, and robust mechanistic-symptomatic associations can be considered to fall under the umbrella of PD. Importantly, we compare clinical data not with imaging features directly, but instead with individualized model weights representing a mixture of imaging and receptor information. Similarly, from the improvements in model residuals due to the inclusion of each receptor distribution, we infer brain regions in which receptor involvement (including direct susceptibility based on receptor expression as well as receptor-pathology interactions) significantly helps explain the accumulation of imaging-measured pathology (Fig. 7, Supplementary Figures S1-S6).

Such use of individualized parameters from neuroimaging models to infer molecular mechanisms of neurodegenerative disorders is a nascent and promising approach (e.g., Stefanovski et al., 2019, <https://doi.org/10.3389/fncom.2019.00054>). Other studies have also found correspondence between personalized model-derived parameters and multi-domain clinical phenotype in Alzheimer's disease (Zimmermann et al., 2018, <https://doi.org/10.1016/j.nicl.2018.04.017>; Khan et al., 2022, <https://doi.org/10.1093/brain/awab375>).

Finally, we confirm that receptor maps from healthy aged subjects have physiological relevance to various neurodegenerative patient populations. Healthy neurotransmitter receptor and transporter templates from PET and SPECT imaging spatially correlate with PD rsfMRI alterations (Dukart et al., 2020, <https://doi.org/10.1002/hbm.25244>), atrophy in multiple sclerosis (Fiore et al., 2023, <https://doi.org/10.1038/s41380-023-01943-1>), brain regions with dyskinesia- and parkinsonism-associated atrophy in schizophrenia patients (Sakrieda et al., 2022, <https://doi.org/10.1093/braincomms/fcac190>), gray matter volume changes in symptomatic and prodromal genetic subtypes of FTD (Premi et al., 2023, <https://doi.org/10.1016/j.nbd.2023.106068>), and functional alterations in behavioral variant FTD (Hahn et al., 2023 - preprint, <https://doi.org/10.1101/2022.08.30.22278624>). Furthermore, in vitro multi-receptor autoradiography of the caudate nucleus and midcingulate area 24 of progressive supranuclear palsy (PSP) patients showed differentiation of autopsied patients from age-matched controls, as well as diverging alterations in clinical subgroups of PSP (Chiu et al., 2017, <https://doi.org/10.1186/s13195-017-0259-5>). In this related movement disorder, notable, previously unknown receptor alterations (to kainate and adenosine type 1 receptors) were

discovered, supporting the case for more thorough receptor studies in neurodegenerative disorders.

Complementing such analyses, in our work, we attempted to use *in silico* analysis to quantify the roles of receptors in mediating specific pathways of interactions between imaging-measured neuropathological processes, regional susceptibility to neurodegeneration, and inter-individual symptomatic variability. Notably, the breadth of pathological factors (i.e., number of imaging metrics) considered in our work is unprecedented in the PD literature, with longitudinal data for each patient to capture temporal progression. Secondly, we present the most comprehensive study of receptor involvement in PD. We characterize how receptors mediate interactions between pathological factors, and, finally, we identify the multivariate associations between mechanistic and symptomatic variability across our cohort of PD patients. Below we address specific points in detail, which have also been incorporated in the Discussion section of the revised manuscript.

Specific aspects:

1. We appreciate the opportunity to clarify the relationships between receptor influence maps, observed gradients of cholinergic loss, and PD pathology. In our models, the combination of multiple types of mechanisms accounts for the longitudinal accumulation of pathology effects (alterations to gray matter density, activity dysfunction, dopaminergic transporter loss, dendrite loss, and directed and undirected microstructural damage). Importantly, templates of healthy receptor distribution are not the only predictors in our models. Receptors contribute to each model by i) directly influencing the accumulation of neuropathology, and ii) mediating the interactions with other forms of neuropathology. In addition, we also model iii) pathological interactions that do not involve receptors, as well as iv) multifactorial network propagation mediated by anatomical connectivity.

Lacking receptor imaging, we instead characterize how various receptors influence neuropathology, either directly or via interactions (e.g., how a patient's regional functional activity and a healthy template of NMDA receptor density together explain regional atrophy). Such model-derived mechanistic pathways are the basis for our results and the features presented in the figures. For example, among the mechanisms contributing to the primary component in Fig. 6a, it is not the distribution of any glutamatergic receptor by itself that predicts neuronal activity alterations, but instead interactions such as kainate receptor densities with regional gray matter density. In other cases, such as the M₁ and M₃ cholinergic receptor effects on undirected microstructure, it is receptor density that helps explain variability in pathological accumulation.

Receptor influence maps are an aggregated metric of all the receptor-affiliated terms acting on a particular neuropathological factor in our models. They specifically show the regions where receptor terms (including direct effects and receptor-pathology interactions) are most informative to each pathological model, above and beyond the non-

receptor terms, across all subjects. Thus, the receptor influence maps are not intended to predict or replicate patterns of neurodegeneration/imaging pathology or receptor loss. It is important to emphasize that they are a “second-order” feature of the effects of receptor density and receptor-mediated interactions on a pathological PD model. We have clarified these points in the description of the Fig. 7 in the subsection *Results: Obtaining PD receptor influence maps*.

2. We appreciate the reviewer giving us the opportunity to clarify the relationships between our receptor influence maps and the other works showing specific gradients of neurotransmission alterations in PD. Firstly, Bohnen et al. (<https://doi.org/10.1093/braincomms/fcac320>) imaged a different molecular target, namely acetylcholinesterase, which has a spatial distribution that is distinct from any of the receptor types included in our work (Supplementary Figs. S1-7). Each (e.g., cholinergic) receptor type, in turn, has a unique distribution, and there is no consistent gradient of healthy receptor distribution among them. These different molecular targets (acetylcholinesterase and the various cholinergic receptors) provide complementary, but not necessarily identical, pictures of the cholinergic system. Secondly, as mentioned in the previous point, each receptor influence map corresponds to a specific receptor-imaging modality pair with aggregated receptor effects from multiple model mechanisms, and does not represent the spatial severity of receptor loss or neuropathological alterations in PD. We have clarified this point in the highlighted changes in the *Discussion* section.
3. We appreciate the reviewer addressing the issue of the limited cortical signal in DAT-SPECT imaging. With the prevalence of DAT-SPECT over DAT-PET in clinical settings, SPECT was the modality of choice in PPMI and the only dopaminergic modality available in a large, multi-center longitudinal imaging study. However, the lack of cortical dopaminergic loss evident in SPECT is a limitation of our data which we have emphasized in the highlighted changes in the revised Discussion section.
We acknowledge the potential sampling bias inherent to the PPMI cohort in combination with our inclusion criteria (at least 3 imaging visits and multiple neuroimaging modalities). We also agree with the reviewer that having control subjects would make straightforward the identification of PD-specific pathology. However, we would like to clarify how the specific methodology of our project bypasses the need for control subjects, to associate model-derived mechanisms with PD. As an inclusion criterion in our work, all subjects were required to have at least 3 imaging and clinical visits to facilitate personalized longitudinal modeling, as well as data for multiple imaging modalities. These criteria disqualified all healthy controls, who generally had less imaging data in the PPMI dataset. Lacking controls, we attempted to identify PD-specific mechanisms by i) identifying multivariate combinations of mechanisms co-varying with PD symptoms, and ii) adjusting for age as a covariate.

PD is primarily defined and diagnosed by clinical symptoms. It is safe to assume that healthy control subjects would not display rapidly deteriorating motor, cognitive, or psychiatric symptoms that are associated with PD. After fitting personalized models individually for each patient, we considered our model-derived mechanisms (obtained from individualized modeling at the patient level) to be associated with PD rather than normal healthy aging when their correlation with multiple PD-related clinical assessments was statistically stable and significant after adjusting for age as a covariable. Thus, the co-variability we find can safely be attributed to the symptomatic spectrum of PD. The multivariate correlation method is described methodologically in the subsection “*Statistical analysis: Covariance of biological mechanisms with clinical symptoms*”, with the results in the subsection “*Two axes of receptor-pathology alterations underlie clinical symptoms in PD*”. This allows us to focus on model-derived mechanisms associated with PD (i.e., clinical symptoms) while accounting for normal (i.e., non-symptomatic) ageing, both within and between subjects. We have emphasized these points in the revised Discussion section.

4. We thank the reviewer for raising the extremely pertinent point about the roles of dopamine and motor symptoms in the primary mechanistic-clinical component. Reflecting the multi-system nature of the disease, our results support the involvement of under-studied pathways and brain regions, while reproducing expected features of PD. Particularly, we would like to emphasize the mainly motor primary axis, and the correspondence between cholinergic mechanisms and visuospatial symptoms of the secondary component. While seemingly diluted by the multiple non-motor symptoms and non-dopaminergic mechanisms, we also note the involvement of the DAT-SPECT alterations in the primary, mainly motor component.

Dopaminergic loss and motor dysfunction are indeed the main pathological and clinical characteristics of PD, which one would naturally expect to feature prominently in the results, and we would like to clarify their contributions to the primary mechanistic-clinical component in our results. These components were obtained by first fitting individual-specific models for each imaging modality, and then performing multivariate (PLS-SVD) comparisons of inter-individual differences in model weights with multiple motor and non-motor clinical symptoms. While D₁ dopaminergic receptor maps were not prominent contributors to this component (Fig. 6a, Supplementary Tables S5 & S7), they are not the only dopaminergic feature in our models. In addition to D₁ dopaminergic maps being predictors, dopamine transporter SPECT imaging is both a model output, and a predictor for itself as well as other imaging modalities. Via both these roles, SPECT does feature prominently in the model-derived mechanisms associated with the primary component. However, as suggested by the reviewer, the lack of cortical SPECT signal may under-emphasize this contribution. The correlation between healthy D₁ distribution and individualized SPECT imaging, as well as its expression patterns in the cortex, also renders the former less informative in neuroimaging models. In addition, while motor

symptoms are the single largest contributor to this primary mechanistic-clinical component (Fig. 5c), there are also prominent contributions of non-motor aspects of experiences of daily living (MDS-UPDRS Part 1; including sleep, urinary dysfunction, pain, psychosis, etc.) and psychomotor speed (the Symbol Digits Modalities Test). As such, the primary mechanistic-clinical axis is a mixed component and does not correspond exclusively to motor symptoms (although they are the most prominent, defining characteristic of this component). Consequently, the dilution of the dopaminergic (including D₁ maps as well as SPECT imaging) mechanism and motor symptom contributions can be expected, as the results reflect the multivariate co-variability of multiple mechanisms (many of which have not been previously characterized in PD) and symptoms rather than the severity of motor symptoms alone. Thus, the dopaminergic system does feature prominently in the mainly motor axis, via the DAT-SPECT model if not the D1 receptor terms, although i) low motor symptom variability, ii) low cortical DAT signal and iii) the mixed symptomatic nature of the component may have underemphasized the classical dopaminergic-motor axis. We have clarified these points in the revised *Discussion* section.

5. We appreciate the reviewer raising the point of non-receptor mechanisms driving pathological accumulation. We emphasize our causal modeling approach, which goes beyond simple correlations to identify pathways of interactions (e.g. to what extent does the interaction between NMDA receptor distribution and gray matter density together explain changes to functional activity, etc.). While the inclusion of receptor maps (and specifically their interactions with individualized neuroimaging) significantly improved the explanation of pathological changes (Fig. 2), reflecting their relevance to explaining PD pathology, it is likely that other important molecular mediators have not been captured. This is potentially reflected in the receptor influence maps, which suggest regional differences in the explanatory benefit of receptor maps in pathological models. For example, this may be due to a molecular feature such as alpha-synuclein, which is an important consideration in the pathophysiology of PD. Radiotracers and imaging markers are only recently emerging for alpha-synuclein, and as a proxy, we did include a propagative model mechanism reflecting the spreading of pathological effects (gray matter atrophy, activity dysfunction, dopaminergic transporter loss, dendrite loss, microstructural damage) within a modality to anatomically connected regions (*Methods: Receptor-Enriched Multifactorial Causal Model*).